# Multiple decisions about one object involve parallel sensory acquisition but time-multiplexed evidence incorporation

Yul HR Kang[1,2†§*], Anne Löffler[1,3†], Danique Jeurissen[1,4†], Ariel Zylberberg[1,5‡], Daniel M Wolpert[1‡], Michael N Shadlen[1,3,4‡*]

[1]Zuckerman Mind Brain Behavior Institute, Department of Neuroscience, Columbia University, New York, United States; [2]Department of Engineering, University of Cambridge, Cambridge, United Kingdom; [3]Kavli Institute for Brain Science, Columbia University, New York, United States; [4]Howard Hughes Medical Institute, Columbia University, New York, United States; [5]Department of Brain and Cognitive Sciences, University of Rochester, Rochester, United States

*For correspondence:
yul.hr.kang@gmail.com (YHRK);
shadlen@columbia.edu (MNS)

[†]These authors contributed equally to this work
[‡]These authors also contributed equally to this work

Present address: [§]Department of Engineering, University of Cambridge, Cambridge, United Kingdom

Competing interests: The authors declare that no competing interests exist.

**Abstract** The brain is capable of processing several streams of information that bear on different aspects of the same problem. Here, we address the problem of making two decisions about one object, by studying difficult perceptual decisions about the color and motion of a dynamic random dot display. We find that the accuracy of one decision is unaffected by the difficulty of the other decision. However, the response times reveal that the two decisions do not form simultaneously. We show that both stimulus dimensions are acquired in parallel for the initial ~0.1 s but are then incorporated serially in time-multiplexed bouts. Thus, there is a bottleneck that precludes updating more than one decision at a time, and a buffer that stores samples of evidence while access to the decision is blocked. We suggest that this bottleneck is responsible for the long timescales of many cognitive operations framed as decisions.

## Introduction

Decisions are often informed by several aspects of a problem, each guided by different sources of information. In many instances, these aspects are combined to support a single judgment. For example, an observer might judge the distance of an animal by combining perspective cues, binocular disparity and motion parallax. In other instances, the aspects are distinct dimensions of the same object. For example, the animal's distance and its identity as potential predator or prey. The former problem of cue combination (*Jacobs, 1999*; *Ernst and Banks, 2002*) is a topic of study in what has been termed the Bayesian Brain (*Knill and Pouget, 2004*). The latter is the subject of this paper. It arises in a wide variety of problems whose solutions depend on identifying a set of conjunctions such as the ingredients of a favorite dish, or when one must make multiple judgments, or decisions, about the same stimulus.

The neuroscience of decision-making has focused largely on perceptual decisions, contrived to promote the integration of noisy evidence over time toward a categorical choice about one stimulus dimension. A well-studied example is a decision about the net direction of motion of randomly moving dots. In such binary decisions (e.g. left or right), behavioral and neural studies have shown that humans and monkeys accumulate noisy samples of evidence and commit to a choice when the accumulated evidence reaches a threshold (*Ratcliff, 1978*; *Palmer et al., 2005*; *Gold and Shadlen, 2007*; *Stine et al., 2020*). The framework has been extended to more than two categories (e.g. *Churchland et al., 2008*; *Bogacz et al., 2007*; *Ditterich, 2010*) but it remains focused on a common stream of evidence bearing on a single stimulus feature. Less is known about how multiple streams

of evidence are accumulated for a multidimensional decision (*Lorteije et al., 2015*). Given the parallel organization of the sensory systems, one might expect all available evidence to be integrated simultaneously. However, there are also reasons to suspect that two decisions cannot be made in parallel. This is based on a variety of experiments that expose a 'psychological refractory period' (PRP; *Welford, 1952*). When participants are asked to make two decisions in a rapid succession, it appears that the second decision is delayed until the first decision is complete (*Pashler, 1994*). Based on such observations, it has been argued that there is a structural bottleneck in the response selection step, such that only one response can be selected at a time (*Sigman and Dehaene, 2005*).

Here, we develop a task in which the participant views one visual stimulus and makes two decisions about the same object. The stimulus comprises elements that give rise to two streams of evidence bearing on their motion and color, and the participant must decide on both aspects and report the combined category. The task was designed to allow participants to integrate both streams of evidence simultaneously from the same location in the visual field and to indicate both choices with just one response. We show that, even in this situation, the two streams of evidence are accumulated one at a time, and moreover, this seriality arises despite the parallel access of the visual system to both streams. We suggest that seriality is explained by a bottleneck between the parallel acquisition of evidence and its incorporation into separate decision processes. We elaborate a model of bounded evidence accumulation, used previously to explain both the speed and accuracy of motion (*Palmer et al., 2005*) and color decisions (*Bakkour et al., 2019*), and show that these accumulations must occur in series. The results have implications for a variety of psychological observations concerning sequential vs. parallel operations, and they address the fundamental question of why mental processes take the time they do.

## Results

We studied variants of a perceptual task that required binary decisions about two properties of a dynamic random dot display. Human participants decided the dominant color and direction of motion in a small patch of dynamic random dots (*Figure 1*). The stimulus is similar to one introduced by *Mante et al., 2013*, who studied the problem of gating when making a decision about only a single dimension, either color or motion. On each video frame, each dot has a probability of being colored blue or yellow and it has another probability of being plotted either at a displacement $\Delta x$ relative to a dot shown 40 ms earlier or, alternatively, at a random location in the display. We refer to the probability of a displacement as the coherence or strength and use its sign to designate the direction. We use an analogous signed probability for the color coherence or strength (see Materials and methods). In the main tasks, participants reported their answer by making an eye or hand movement to select one of four choice targets. We refer to this as a double-decision and refer to the two aspects as stimulus dimensions. We employed several variants of this basic task in our study.

### Roadmap of the experimental results

We first present the main finding using a free response paradigm, what we term *double-decision reaction time* (Experiment 1). It demonstrates no interference in choice accuracy—that is, the difficulty of the color decision does not affect the accuracy of motion decisions, and vice versa—but critically, the double-decision (2D) time is the sum of the two single-decision (1D) times. The analysis suggests that the motion and color decisions are not formed at the same time. This establishes the prediction that with brief stimulus presentations, successful color decisions ought to be attained at the expense of motion, and vice versa—that is, choice interference. We then test this prediction (Experiment 2) and fail to confirm it. We show that color and motion can be acquired in parallel but are unable to update the decision simultaneously. This confirms the response selection bottleneck predicted by Pashler (*Fagot and Pashler, 1992*) and it implies the existence of buffers (*Sperling, 1960*; *Kamienkowski and Sigman, 2008*), where sensory information can be held before it updates a decision variable—the accumulated evidence for color or motion.

The combination of a buffer and serial updating leads to a revised prediction that interference in accuracy should occur over a narrow range of stimulus viewing duration, controlled by the experimenter. We confirm this prediction (Experiment 3), showing that there is no interference at short viewing times, but that there is a narrow regime of the stimulus duration in which accuracy on one dimension suffers because a limited amount of deliberation time needs to be shared with the other

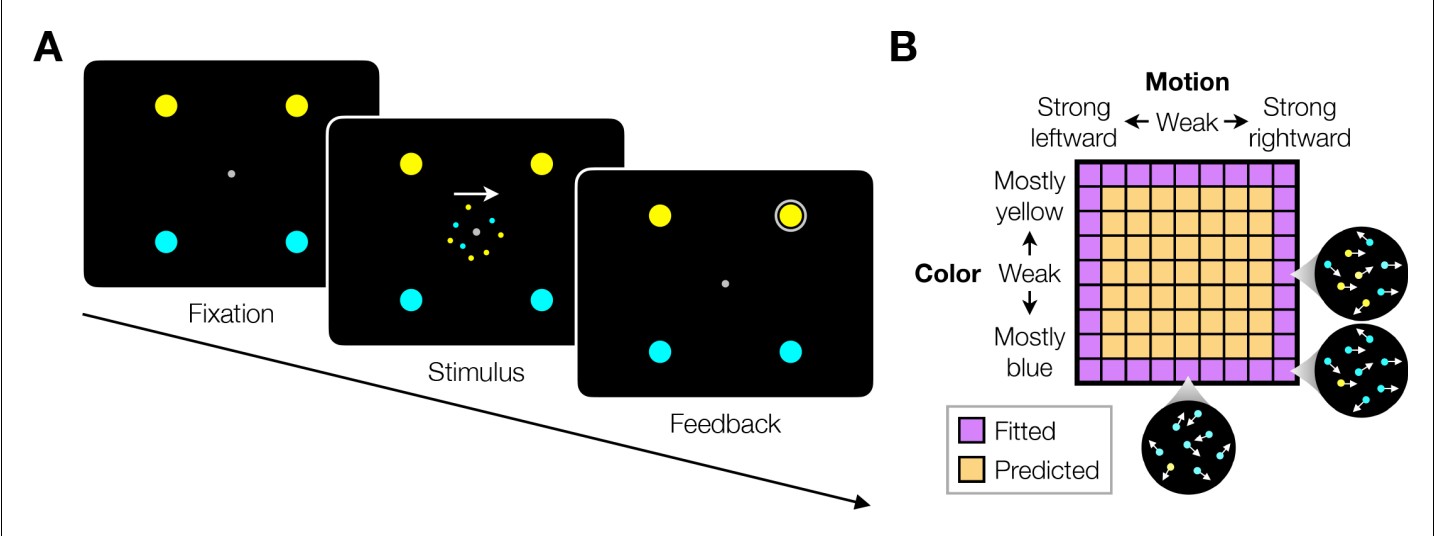

**Figure 1.** Double decision task. (A) Timeline of the behavioral task. Participants fixated a gray dot at the center of the screen. A dynamic random dot stimulus was displayed and the participant was asked to judge the overall motion direction and the dominant color (the arrow is for visualization purposes only and was not presented to the subject). They reported this double-decision by selecting one of four targets to indicate motion direction (left and right target for leftward and rightward motion, respectively) and color (top yellow vs. bottom blue targets). The response was deemed correct when both motion and color judgments were correct. Participants received auditory feedback as to whether they were correct and the correct target was also indicated by a white ring. Across the experiments the targets could be selected with an eye movement or a hand movement, either when the participant was ready to report (reaction time) or when the dot display was extinguished (experimenter-controlled duration). (B) Motion and color strengths were varied independently across trials, represented by a matrix of combinations of difficulty levels (here shown for the eye reaction-time experiment with 81 combinations; see Materials and methods for Exp. 1-eye). Insets illustrate typical motion and color for three of the conditions. For feedback only, choices on the weakest motion strength (0% coherence) were deemed correct randomly; same for the weakest color strength. For the combinations shown in purple, at least one stimulus dimension was at its strongest value (easiest). For some analyses, the data from these combinations are used to fit a model, which is evaluated by predicting the data from the remaining combinations (amber).

dimension, which reconciles conflicting observations of parallel and serial patterns of decision-making in the literature (e.g. *Schumacher et al., 2001*; *Tombu and Jolicoeur, 2004*). We then introduce a bimanual version of the task (Experiment 4) which affords direct reports of both the color and motion termination times. It confirms the assumption that the double-decision time is the sum of two sequential sampling processes, each with its own stopping time, and it shows that the color and motion decisions compete before the first decision terminates. This implies some form of time-multi-plexed alternation. In the last experiment, we ask participants to judge whether the motion in a pair of patches is the same or different (Experiment 5) and find that this binary decision, based only on motion processing, also exhibits additive decision times. Finally, we introduce a conceptual model of the double-decision process that serves as a platform to connect the computational elements with known and unknown neural mechanisms.

## Experiment 1. Double-decision reaction time (eye and unimanual)

Participants were asked to judge both the net direction (left or right) and dominant color (yellow or blue) of a patch of dynamic random dots and to indicate both decisions with a single movement to one of four choice targets (*Figure 1A*). Different groups of participants performed the task by indicating their choices with an eye movement (Exp. 1-eye, N = 3) or a reach (Exp. 1-unimanual, N = 8; see Figure 5A). On each trial the strength and direction of motion as well as the strength and sign of color dominance were chosen independently, leading to 81 (9 × 9 in Exp. 1-eye) or 121 (11 × 11 in Exp. 1-unimanual) combinations. The single movement furnished two decisions and one response time (RT; n.b., We use the terms, response time and reaction time, interchangeably to respect usage in psychology and neurophysiology literatures). Participants were given feedback that the decision was correct if the motion and color were both correct (see Materials and methods). After initial training, each participant in Exp. 1-eye performed 4,624–10,969 trials over 11–17 sessions; each participant in Exp. 1-unimanual performed 2304 trials over two sessions.

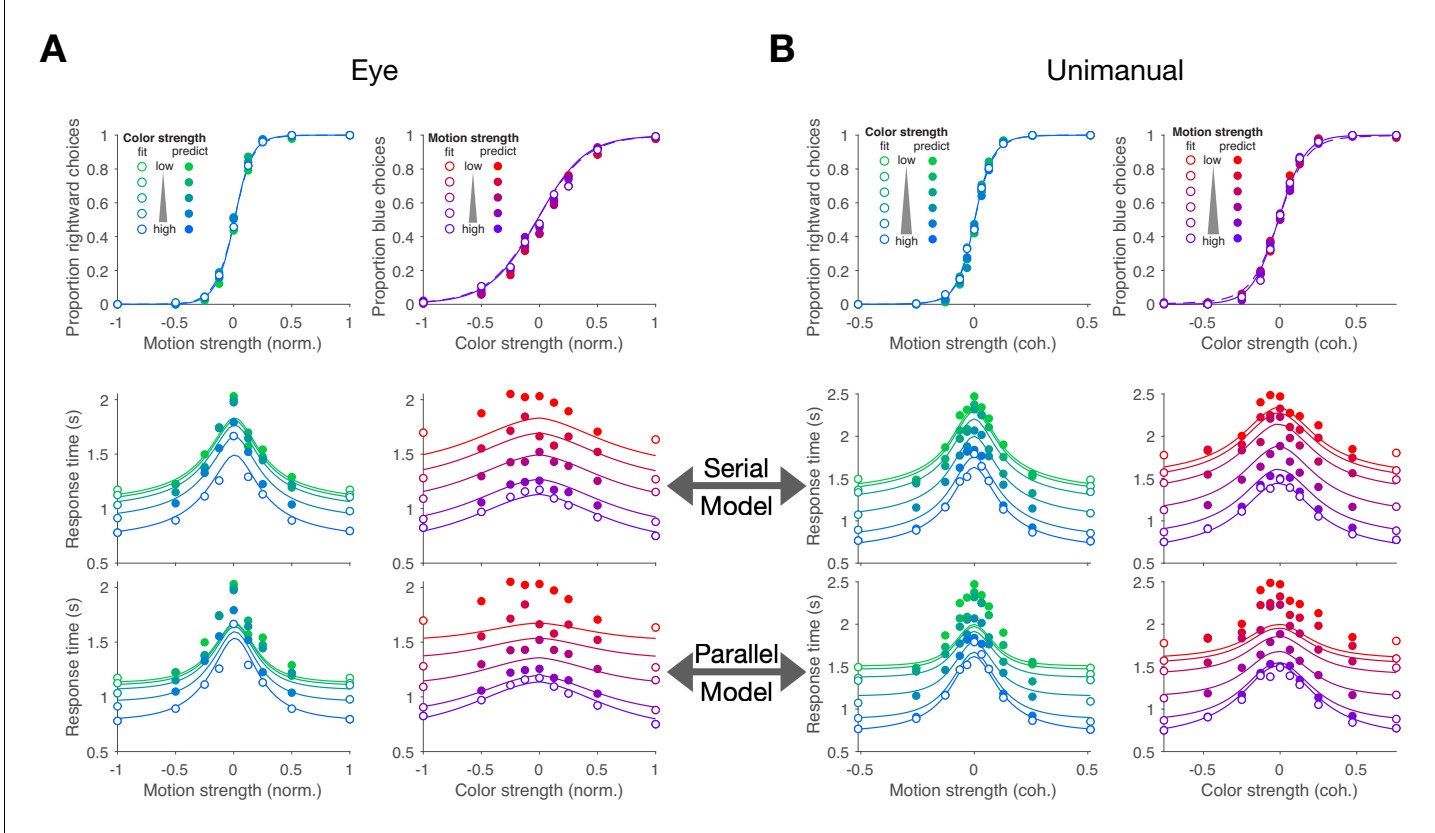

**Figure 2.** Double-decisions exhibit additive response times but no interference in accuracy (Experiment 1). Participants judged the dominant color and direction of dynamic random dots and indicated the double-decision by an eye movement (**A**; Exp. 1-eye) or reach (**B**; Exp. 1-unimanual) to one of four choice-targets. All graphs show the behavioral measure (proportion of choices, top row; mean RT, rows 2 and 3) as a function of either signed motion or color strength. Positive and negative color strength indicate blue- or yellow-dominance, respectively. Positive and negative motion strength indicate rightward or leftward, respectively. Colors of symbols and traces indicate the difficulty (unsigned coherence) of the other stimulus dimension (e.g., color, for the graphs with abscissae labeled 'Motion strength'). Symbols are combined data from three participants (Exp. 1-eye) and eight participants (Exp. 1-unimanual). Open symbols identify the conditions used to fit the serial (middle row) and parallel (bottom row) models. These are the conditions in which at least one of the two stimulus strengths was at its maximum (purple shading, *Figure 1B*). In the top row, fits of the serial and parallel models are shown by solid and dashed lines, respectively. The models comprise two bounded drift-diffusion processes, which explain the choices and decision times as a function of either color or motion. They differ only in the way they combine the decision times to explain the double-decision RT. For the serial model, the double-decision time is the sum of the color and motion decision times. For the parallel model, the double-decision time is the longer of the color and motion decisions (see Materials and methods). Smooth curves are the predictions based on the fits to the open symbols. Both models predict no interaction on choice (top row). The predictions of RT are superior for the serial model (middle row) compared to the parallel model (bottom row). Data are the same in the lower two rows. Stimulus strengths in A were not identical for the three participants and were normalized to a common ±1 scale before averaging, so the psychometric curves for eye and hand cannot be compared visually (see *Appendix 1—table 1* for comparison of parameters from the fits). For simplicity, only correct (and all 0% coherence trials) are shown in the RT graphs (see Materials and methods).

The online version of this article includes the following figure supplement(s) for figure 2:

**Figure supplement 1.** Statistical comparison of the drift diffusion model under serial vs. parallel rules (Experiments 1 and 4).

**Figure supplement 2.** Comparison of parallel and serial rules applied to RT distributions (Experiments 1 and 4).

**Figure supplement 3.** Statistical comparison of parallel and serial rules applied to reaction time distributions (Experiments 1, 4, and 5).

**Figure supplement 4.** Mean reaction time for parallel and serial rules applied to the empirical analysis of reaction time distributions exemplified in Figure 2—figure supplement 2 (Experiments 1 and 4).

**Figure supplement 5.** Application of the fit-prediction strategy in *Figure 2* using only reaction time distributions.

**Figure supplement 6.** Sensitivity of color and motion choices on single-decision and double-decision tasks (Exp. 1-eye).

*Figure 2A,B* shows choices and mean RT as a function of stimulus strength for the eye and unimanual tasks, respectively. The graphs in the left column of each panel show the data plotted as a function of motion strength and direction. Each color on this graph corresponds to a different difficulty

of the other dimension (i.e. the color decision). Similarly, the graphs in the right columns show the data plotted as a function of color strength and dominance; the strength of the uninformative dimension, motion, is shown by the purple/red shading. Unsurprisingly, the proportion of rightward choices increased as a function of the sign and strength of the motion coherence, and the proportion of blue choices increased as a function of the sign and strength of color coherence. The striking feature of these graphs is that sensitivity to variation in the stimulus along each dimension is unaffected by the difficulty along the uninformative dimension. This is evident from the superposition of the colored data points. It is also supported by a logistic regression analysis, which favored a choice model in which the sensitivity along one dimension is not influenced by the stimulus strength along the other dimension ($\Delta$BIC = 23 and 22 for motion and color in the eye task, respectively; $\Delta$BIC = 34 and 52 for the unimanual task; positive values are support for the regression model of *Equation 13* without the $\beta_3$ term). It implies that the two stimulus dimensions do not interfere with each other. This is consistent with the well-established idea that color and motion are processed by parallel, independent channels (*Livingstone and Hubel, 1988*; *Ramachandran and Gregory, 1978*; *Carney et al., 1987*; *Cavanagh et al., 1984*). However, another possibility is that the two dimensions do not interfere because they are not processed simultaneously but serially.

The RTs support this serial hypothesis. The RTs, plotted as a function of either motion or color, are bell-shaped curves, such that longer RTs are associated with the most difficult stimulus strength and the fastest with the easiest. In contrast to the choice functions, the uninformative dimension—that is, with respect to the dimension of the abscissa—affects the scale of these RTs, giving rise to a stacked family of bell-shaped curves. The more difficult the other dimension, the longer the RT.

We attempted to explain the choice-RT data in *Figure 2* with models of bounded evidence integration (e.g. drift-diffusion; *Ratcliff, 1978*; *Palmer et al., 2005*). Such models provide excellent accounts of choice and RT on the motion-only and color-only versions of these tasks (*Palmer et al., 2005*; *Bakkour et al., 2019*). To explain the double-decision data set, we pursued two variants of these models under the assumption that motion and color are processed in parallel or in series. The curves in *Figure 2* are a mixture of fits and predictions. To fit the data, we used all trials in which at least one of the dimensions was at its strongest level (open symbols *Figure 2*; 32 purple conditions in *Figure 1B* for the eye task and 40 conditions for the unimanual task). We used these fits to predict the data from the remaining conditions (filled symbols; 49 amber conditions in *Figure 1B* for the eye task; 81 for the unimanual task).

Both models are consistent with no interference in the choice functions. The models can be distinguished on the basis of the RT data. For an experiment with only a single dimension (e.g. motion), the RT is the sum of the amount of time that evidence is integrated to reach a terminating bound (the decision time, $T_\mathrm{m}$ or $T_\mathrm{c}$, for motion and color choice, respectively) plus time delays that are not affected by task difficulty, such as sensory and motor delays, termed the non-decision time ($T_\mathrm{nd}$). If the color and motion decisions are made in parallel, then the total decision time should be determined by the slower process ($\max[T_\mathrm{m}, T_\mathrm{c}]$), whereas if the decisions are made serially, the total decision time would be determined by the sum of the two decision times ($T_\mathrm{m} + T_\mathrm{c}$). In both cases, we expect both motion and color strengths to affect the RT. In the serial case, an increase in the difficulty of color, say, should augment the total RT by the same amount for all motion strengths, giving rise to stacked functions of the same shapes (solid curves, middle row, *Figure 2A,B*). In the parallel case, an increase in the difficulty of color should augment the total RT by an amount that depends on the difficulty of motion (solid curves, bottom row *Figure 2A,B*). The color dimension is likely to determine the total RT when motion is strong, but it has less control when the motion is weak. The logic should produce stacked bell-shaped functions that pinch together in the middle of the graph. The data are better explained by the serial predictions (e.g. large mismatches by the parallel model when both dimensions are weak). Formal model comparison provides strong support for the serial models overall (geometric mean of Bayes factor across participant and task combinations: $\log_{10} \mathrm{BF} > 66$) and for 9 of 11 participants individually (*Figure 2—figure supplement 1*).

The systematic underestimate of RT by both models on the doubly difficult stimulus strengths arises for two reasons. First, the drift-diffusion model implements a restrictive assumption that the variance of the noisy momentary evidence is the same for all stimulus strengths. The variance of the momentary evidence is likely to be smaller near 0% (see Materials and methods). The overestimate of the variance in the model would lead to an underestimate of the RT in the middle of the graphs of RT vs. motion strength, and along the top of the graphs of RT vs. color strength. Second, the

inclusion of all trials at 0% coherence tends to inflate the mean RT because just under half of the trials resemble errors in the sense that the choice is opposite the sign of the component of the drift rate that instantiates a direction or color bias (e.g. $s_m^0 \neq 0$, *Equation 2*). Importantly, we pursued a second approach to compare serial and parallel models which shows that the superiority of the serial model does not rest on the systematic underestimates of RT in *Figure 2*.

In the second approach, we focus specifically on the decision times. It considers only the distribution of RTs and attempts to account for them under serial and parallel logic. Instead of fitting diffusion models, this *empirical* approach explains the observed double-decision RT distributions as either the serial or parallel combination of latent (i.e. unobservable) distributions of color and motion decision times, as well as the four $T_{nd}$ distributions (one for each choice). We estimate these latent distributions with gamma distributions. For the serial case, the predicted double-decision RT distributions are established by convolution of the latent single-dimension distributions and the distribution of $T_{nd}$. For the parallel case, the latent distributions are combined using the max logic, and the result is convolved with the appropriate distribution of $T_{nd}$ (see Materials and methods). *Figure 2—figure supplement 2* shows fits to the double-decision RT distributions for the more informative conditions for the serial and parallel models. The model comparisons, based on all the data, yield 'decisive' support (*Kass and Raftery, 1995*) for the serial processing of motion and color (geometric mean of Bayes factor for participant and task combinations $\log_{10} BF > 17$ with all but one out of 11 participants individually supporting the serial rule; *Figure 2—figure supplement 3*). We also display the mean RTs derived from the fits in the same format as *Figure 2* (*Figure 2—figure supplement 4* and *Figure 2—figure supplement 5*). Both approaches support the conclusion that the color-motion double-decisions are formed serially from two independent decision processes, each with its own termination rule. However, neither analysis discerns the nature of the serial processing (e.g. whether they alternate or one is prioritized). We will consider this issue later.

## Experiment 2. Brief stimulus presentation (eye)

The results from the double-decision RT experiment support sequential updating of two decision variables, which represent accumulated evidence for the motion and color choices. If this is true, it leads to a straightforward prediction. If the stimulus duration is not controlled by the decision maker but by the experimenter, and if it is brief, then the two stimulus dimensions would compete for the limited processing time, and we ought to observe choice-interference. We therefore conducted a second experiment with the participants from Exp. 1-eye (N = 3). In this experiment, we presented the same motion/color coherence combinations, but limited the duration of the stimulus viewing time to just 120 ms. We know from previous experiments with 1D tasks that performance continues to increase with stimulus duration up to at least one half second (*Kiani et al., 2008*; *Waskom and Kiani, 2018*). Thus, it is reasonable to assume that performance accuracy would suffer if it is not possible to make use of the full 120 ms of evidence for both motion and color. We predicted that sensitivity to both color and motion should be worse on the double-decision task than on color-only and motion-only versions of the identical task. Each participant performed a total of 7305–7741 trials (4052–4275 1D trials and 3240–3466 2D trials) over 12–19 days.

To our surprise, double-decisions were just as accurate as their 1D controls (*Figure 3A*). We also observed no change in the sensitivity to color across the range of motion difficulties, and vice versa ($\Delta$BIC = 7.2 and 9.7 for motion and color choices, respectively, in support of no interaction; *Equation 12*, $\beta_3 = 0$). This suggests that evidence for color and motion was acquired simultaneously, in parallel, and without interference. Further support for this conclusion is adduced from an analysis of the stimulus information used to make the decisions—what is known as psychophysical reverse correlation (*Beard and Ahumada, 1998*; *Okazawa et al., 2018*). *Figure 3B* displays the degree to which trial-by-trial variation in the noisy displays influences the choice (see Materials and methods). It shows that these stimulus fluctuations influenced choices almost identically in the double-decision task and 1D controls.

At first glance, the observation seems to be at odds with our interpretation of the double-decision RT experiment, which provided strong support for serial processing, primarily in the pattern of RTs. In Experiment 2, the entire stimulus stream lasts only 120 ms, which is less than a typical saccadic latency to a bright spot. Nevertheless, participants exhibited variation in the time of their responses as a function of stimulus strength (*Figure 3A*, bottom panels) and these RTs were

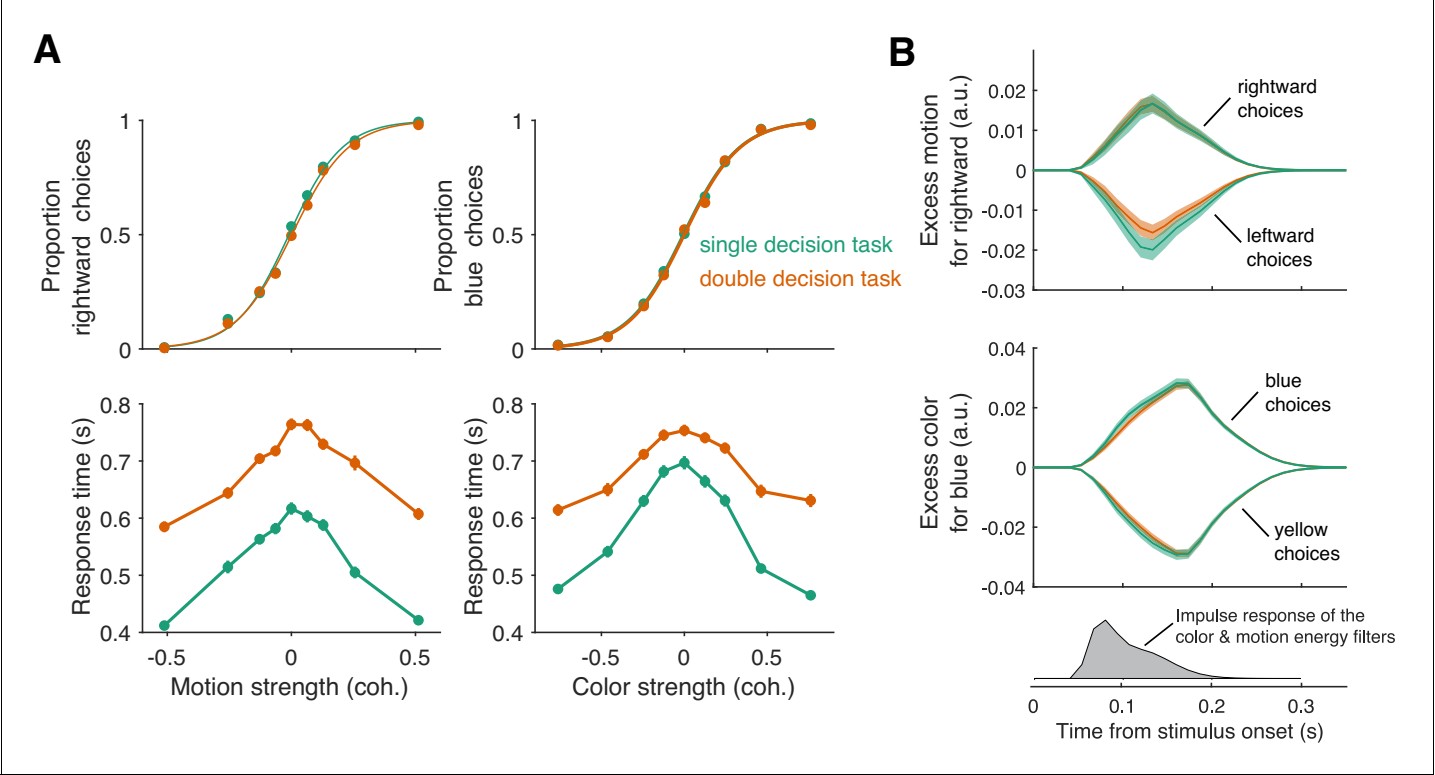

**Figure 3.** Parallel acquisition and serial incorporation of a brief color-motion pulse (Experiment 2). Participants completed a short-duration variant of the double-decision task in which the stimulus was presented for only 120 ms. They also performed blocks in which they were asked to report only the color or only the motion direction (single decision in which they could ignore the irrelevant dimension). Data from double- and single-decision blocks are indicated by color. (**A**) Choices and RTs for single and double-decision blocks. *Top-left*, proportion of rightward choices as a function of motion strength. *Top-right*, proportion of blue choices as a function of color strength. The solid lines are logistic fits. They are nearly identical for single- and double-decisions. *Bottom row*, RT for the single- and double-decisions plotted as a function of motion strength (*left*) and color strength (*right*). For double-decisions, these are the same data plotted as a function of either the motion or color dimension. Data points show the average RT as a function of motion or color coherence, after grouping trials across participants and all strengths of the 'other' dimension (i.e. color, *left*; motion, *right*). Error bars indicate s.e.m. across trials. Although the stimulus was presented for only 120 ms, RTs were modulated by decision difficulty. Importantly, RTs were longer in the double-decision task than in the single-decision task. (**B**) Psychophysical reverse correlation analysis. *Top*, Time course of the motion information favoring rightward, extracted from the random-dot display on each trial, that gave rise to a left or right choice. Shading indicates s.e.m. *Middle*, Time course of the color information favoring blue, extracted from the random-dot display on each trial, that gave rise to a blue or yellow choice. Shading indicates s.e.m. The similarity of the green and orange curves indicates that participants were able to extract the same amount of information from the stimulus when making single- and double-decisions. *Bottom*, Impulse response of the filters used to extract the motion and color signals (see Materials and methods). They explain the long time course of the traces for the 120 ms duration pulse.

surprisingly long. The fastest were ~300 ms longer than the stimulus (RT>400 ms). Importantly, they are approximately 100–200 ms longer in the double-decisions than in single decisions. It is difficult to make too much of this observation, because the participants might have procrastinated for reasons unrelated to the dynamics of the decision process. However, procrastination would not explain the difference between the two conditions. As parallel acquisition of the 120 ms color and motion take the same amount of time as acquisition of either of the streams alone (by definition), the extra time in the double-decision is probably explained by serial incorporation of evidence into the two decisions. This observation also implies the existence of buffers that store the information from one stream as it awaits incorporation into the decision.

Our results so far suggest that color and motion information are acquired in parallel but are incorporated into the decision in series. We therefore wondered if the same schema might apply to the double-decision RT task. For this to hold, some kind of alternation must occur such that segments of one or the other stimulus stream is not incorporated into its decision. Suppose, for example, that at $t = 120$ ms, motion information had been incorporated into decision variable, $V_m$, and color information had been stored in a buffer. Suppose further that motion continues to update the decision

variable, $V_\mathrm{m}$, until it reaches a termination bound at $t = T_\mathrm{m}$, and only then can the buffered color information be incorporated into decision variable, $V_\mathrm{c}$. From then on color information could update $V_\mathrm{c}$ until this decision terminates. In this imagined scenario, the color information between 0.12 s and $T_\mathrm{m}$ is not incorporated in the decision.

One might also imagine two alternatives to the latter part of this scenario. In both, the information from color continues to update the buffer (but not $V_\mathrm{c}$) throughout the motion decision without loss. Then at $t = T_\mathrm{m}$ either (i) all the information about color is incorporated immediately into $V_\mathrm{c}$ or (ii) the buffered information is incorporated in $V_\mathrm{c}$ over time (e.g. as if the recorded color information is played back). The first alternative is equivalent to the parallel model that is inconsistent with the data. The second alternative, implausible as it may seem, implies the color decision is blind to the color information in the display during the playback of the recorded color information. These alternatives are not intended as serious models but to convey two general intuitions. First, if there is a buffer at play in the double-decision RT task then it must take time for the buffered information to be incorporated, or the RTs would have conformed to the parallel logic. Second, if the duration of the buffer is finite, when both 1D processes require more processing time than the duration of the buffer, there will be portions of the color and/or motion stimulus that do not affect the decision.

One might therefore ask why the second point does not lead to a reduction in sensitivity (or accuracy) in color, say, when motion is weak and competes with color for processing time. The answer is that when the decision maker controls the termination of the decision, they can compensate for the missing information by collecting more, until the level reaches the same terminating bound. This leads to a straightforward prediction. If the experimenter controls the termination of the evidence stream, then missing portions of the color and/or motion stimulus might impair performance, especially when the other stimulus dimension is weak.

## Experiment 3. Variable-duration stimulus presentation (eye)

We therefore predicted that under conditions in which the experimenter controls the viewing duration, there is an intermediate range of viewing durations, greater than 120 ms and less than the average RT of difficult double-decisions, where we might observe interference in sensitivity. To appreciate this prediction, it is essential to recognize that when the experimenter controls viewing duration of a random dot display, the decision maker applies a termination criterion, as they do in choice-RT experiments (*Kiani et al., 2008*). There is no overt manifestation of this termination, although it can be identified by introducing perturbations to the stimulus (see also *Kang et al., 2017*). Before such termination, sensitivity improves by the square root of the stimulus viewing duration ($\sqrt{t}$) as expected for perfect integration of signal-plus-noise. In a double-decision, when the two decision processes are splitting the time equally, the sensitivity of each should only improve by $\sqrt{t/2}$. However, when one process terminates, the rate of improvement of the other process should recover, until that process reaches its terminating bound. The model predicts a range of stimulus strengths and viewing durations in which interference in accuracy ought to be evident. It also predicts that the range and degree of interference might depend on which stimulus dimension the participant prioritizes. Here, we set out to test this prediction.

Two participants each performed ~11,800 trials over 12–16 sessions. The task was identical in structure to the brief-duration experiment. However, stimuli were presented at fixed durations ranging from 120 to 1200 ms (in steps of 120 ms). Only three levels of difficulty were used for each dimension: one easy and two difficult coherence levels. The two difficult coherence levels were adjusted individually to yield 80% and 65% accuracy, respectively, ensuring above-chance performance despite the high difficulty level. The easy coherence level was fixed at the highest motion/color coherence from Experiment 1-unimanual, as this coherence typically supports perfect accuracy. The number of coherence levels was reduced compared to Experiments 1 and 2 in light of the large number of conditions (6 signed motion coherences × 6 signed color coherences × 10 stimulus durations). The key comparison here is sensitivity to difficult color, say, when (i) motion is difficult and therefore likely to compete with the color for decision time vs. (ii) motion is easy and less likely to wrest time away from color. This comparison within the double-decision task is more appropriate than a comparison between the double and single decision tasks as these tasks are likely to elicit different termination bounds, as they have different error rates—0.75 and 0.5—on difficult trials (see Materials and methods).

*Figure 4* shows the sensitivity to motion and color as a function of stimulus duration, when the other stimulus dimension was easy or difficult. The sensitivity is the slope of a logistic fit of the motion (or color) choices to the three levels of difficulty. Notice that for both participants there is no difference in sensitivity at the shortest stimulus duration (120 ms), consistent with the findings above. However both participants exhibited lower sensitivity at intermediate durations when color choices were coupled with difficult motion. This difference implies an interference. It is less compelling, if present at all, when motion choices are coupled with difficult color. This pattern, in which motion difficulty affects color sensitivity but not vice-versa, is consistent with participants prioritizing one decision over the other. This would arise if participants consistently monitored the motion stream first and turned to color after the motion decision terminated. In this case, the difficulty of the color would not affect the decisions for motion, but harder motion would take longer to terminate, thereby leaving less time for color processing. We therefore fit a model in which one decision was prioritized over another by including a parameter that determined the probability that motion would be processed first. We also included a parameter that controls the duration of the stimulus streams that can be held in the buffer. This is, effectively, the amount of stimulus information that can be acquired in parallel. The best fits of the model, shown by the smooth curves (*Figure 4A*), suggest a buffer capacity of approximately 80 ms worth of stimulus information (*Figure 4B* and *Figure 4—figure supplement 1*) and prioritization of motion on approximately 80–96% of trials. The serial and parallel models are special cases of this model in which the buffer durations are very short or long, respectively. Both such buffer capacities provide very poor fits to the data (*Figure 4—figure supplement 2*). Note that we continue to refer to the model as serial because even in the parallel phase, the decision is updated serially.

The findings therefore support our prediction, and in doing so, they support the hypothesis that a common principle underlies double-decisions ranging from a tenth to at least two seconds, independent of whether this duration is controlled by the experimenter or by the decision maker. Namely, there is parallel acquisition but serial incorporation of color and motion into the double-decision process. The interference in choice accuracy demonstrated in this experiment is the only example of choice interference in our study. It is remarkably elusive, because it can be observed only for stimulus durations for which three conditions are satisfied: (i) the duration of the stimulus is long enough that parallel acquisition is no longer possible; (ii) the duration of the stimulus is short enough that accuracy on one dimension would benefit from additional sensory evidence; (iii) the duration should support termination of the other dimension for strong but not weak stimuli. The interference is also deceptive. It is explained by a competition for processing time, not by an interaction affecting the fidelity of the sensory streams themselves. It is an example of resource sharing (*Tombu and Jolicoeur, 2002*; *Tombu and Jolicoeur, 2005*; *Kahneman, 1973*), but the resource is time, specifically.

## Experiment 4. Two-effector double-decision reaction time (bimanual)

There are two important features of the serial model: the existence of two decision variables that are terminated independently, and that these accumulations are not updated at the same time but in series. A limitation in the experiments so far is that we had access to the completion of the double-decision but not to the completion of each component. Therefore, we could only speculate about which decision completed first, and when. Without knowledge of the first decision time, we cannot tell how often a participant switched between updating the motion and color decision variables. For example, the prioritization considered in the previous section could arise by completing one decision before deliberating on the second or by alternating back and forth on a schedule that allocates more time to motion. We therefore conducted an experiment in which participants indicated their choice and RT for each stimulus dimension using separate effectors.

The eight participants who performed the unimanual version of the double-decision RT task (Exp. 1-unimanual) also performed a bimanual version of the same task (*Figure 5A*). In the unimanual version, participants used a handle to move a cursor to one of four targets that simultaneously communicated color and motion decisions. In the bimanual version, participants indicated their motion decision by moving one of the handles in a left/right direction and indicated their color decision with a forward/backward movement of the other handle. Participants were instructed to report each 1D decision as soon as it was made. To facilitate this, they received extensive training, consisting of blocks in which one of the stimulus dimensions was set at its easiest level. Both the order of the tasks (unimanual and bimanual) and the hand assignments (left/right × color/motion) were balanced

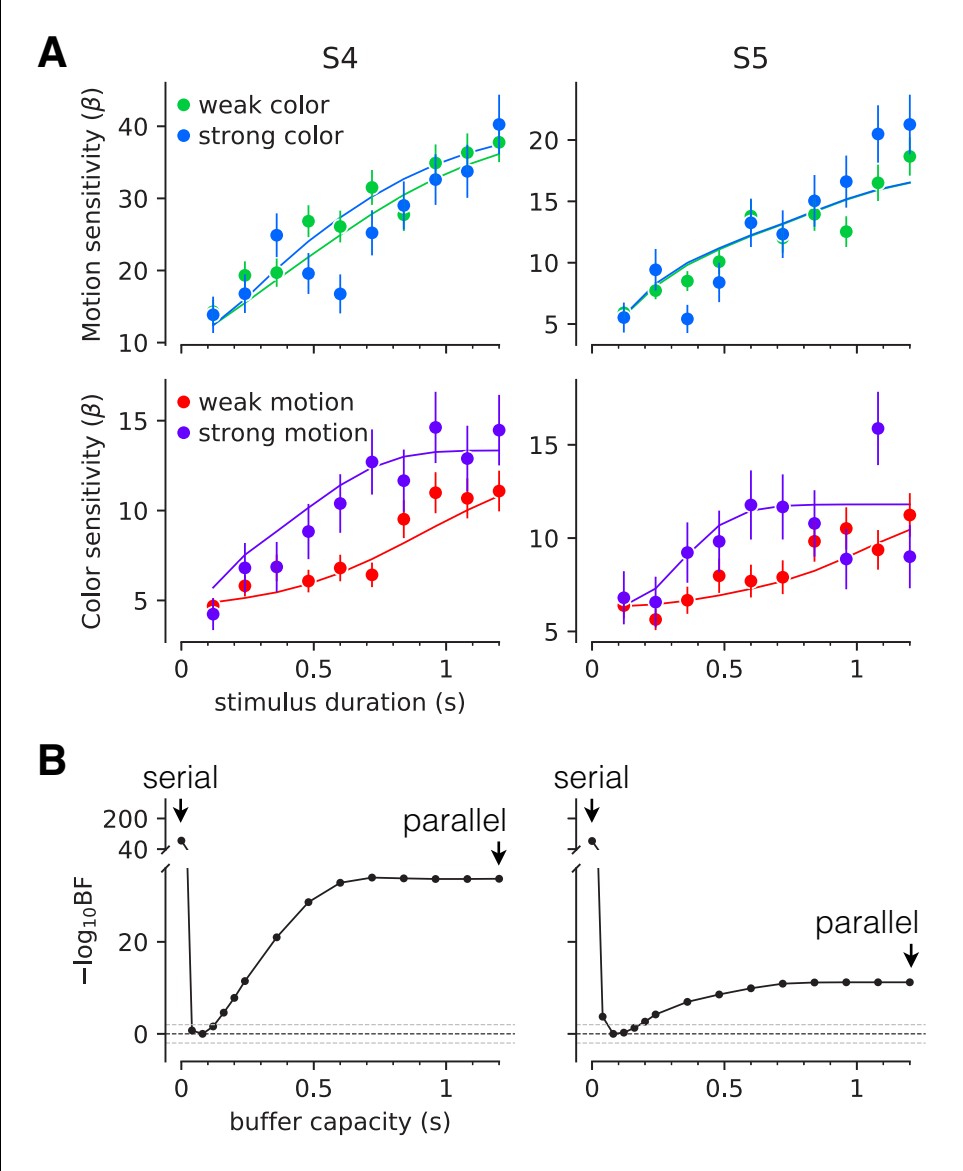

**Figure 4.** Interference in choice accuracy can be elicited at intermediate viewing durations (Experiment 3). Two participants (columns) performed the color-motion double-decision task with a random dot display presented for 120–1200 ms. (**A**) *Top*, Motion sensitivity as a function of stimulus duration and color strength. Symbols are the slope of a logistic fit of the proportion of rightward choices as a function of signed motion strength, for each stimulus duration. Data are split by whether the color strength was strong (blue) or weak (green). Error bars are s.e. *Bottom*, Analogous color-sensitivity split by whether the motion strength was strong (purple) or weak (red). Curves are fits to the data from each participant using two bounded drift diffusion models that operate serially after an initial stage of parallel acquisition, here termed the buffer capacity. During the serial phase, one of the dimensions is prioritized until it terminates. The prioritization favored motion for both participants ($p_{\mathrm{m}}^{\mathrm{1st}} = 0.80$ and 0.96, for participants S4 and S5, respectively). (**B**) Negative log likelihood of the model fits as a function of the buffer capacity, relative to the model fit at 80 ms capacity. The model is equivalent to a purely serial model, when the buffer capacity is zero, and to a purely parallel model when the buffer capacity exceeds the maximum stimulus duration. Negative log likelihoods were computed for a discrete set of buffer capacities (black points). Horizontal lines at $\log_{10} \mathrm{BF} = 0$ indicate Bayes factor = 1. Dashed lines show where the Bayes factor = ± 100 ('decisive' evidence; *Kass and Raftery, 1995*).

The online version of this article includes the following figure supplement(s) for figure 4:

**Figure supplement 1.** Parameter recovery analysis (Experiment 3).

**Figure supplement 2.** Fits to the choice data in Experiment 3 with strictly serial and parallel models.

between the participants (see Materials and methods). Trial numbers (2304) and motion-color coherence levels (11 × 11 combinations of signed coherence levels) were identical for the uni- and bimanual version of the task.

Before tackling the questions that motivate the bimanual experiment, we first ascertained whether participants used the same strategy to make bimanual double-decisions as they did on the unimanual version. It seemed conceivable that by using separate hands to indicate the motion and color decisions, participants could achieve parallel decision formation, for example, as a pianist reads the treble and bass staves with the right and left hands, typically. We therefore conducted a model comparison similar to that of *Figure 2*. To fit the models, we used the color and motion choice on each trial along with the second response time ($RT^{2nd}$) regardless of whether it was to indicate

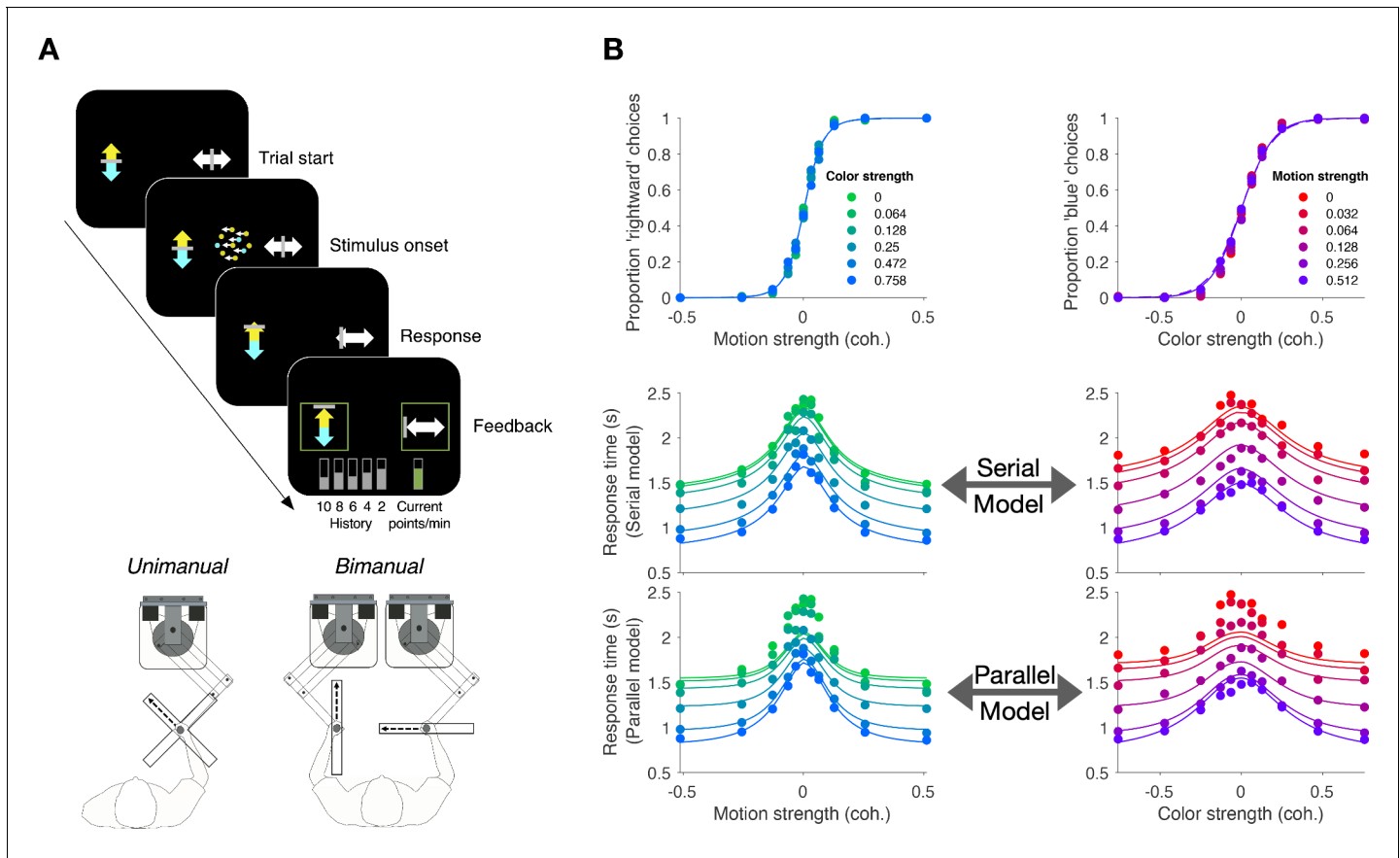

**Figure 5.** Replication of double-decision choice-reaction time when the decisions are reported with two effectors (Experiment 4). (**A**) Participants performed the color-motion double-decision choice-reaction task, but indicated the double-decision with either a unimanual movement to one of four choice-targets or a bimanual movement in which each hand reports one of the stimulus dimensions (N = 8 participants performed both tasks in a counterbalanced order). In both conditions, the hand or hands were constrained by a robotic interface to move only in directions relevant for choice (rectangular channels). The display was the same in the unimanual and bimanual tasks, with up-down movement reflecting color choice and left-right movement reflecting motion choice. A scrolling display of proportion correct was used to encourage accuracy. In the unimanual trials both choices were indicated simultaneously. However, in the bimanual trials each choice could be indicated separately and the dot display disappeared only when the second hand left the home position. (**B**) Choice proportions and double-decision mean RT on the bimanual task. The double-decision RT on the bimanual task is the latter of the two hand movements. The data are plotted as a function of either signed motion or color strength (abscissae), with the other dimension shown by color (same conventions as in *Figure 2*). Solid traces are identical to the ones shown in *Figure 2B* for the unimanual task, generated by the method of fitting the conditions containing at least one stimulus condition at its maximum strength and predicting the rest of the data. They establish predictions for the bimanual data from the same participants. The agreement supports the conclusion that the participants used the same strategy to solve the bimanual and unimanual versions of the task. Note that a few symbols are occluded by others.

The online version of this article includes the following figure supplement(s) for figure 5:

**Figure supplement 1.** Choice and double-decision RT for the bimanual responses (Experiment 4) in the same format as *Figure 2B*.

**Figure supplement 2.** Model-free comparison of performance in the unimanual (blue) vs. bimanual (red) task (Experiment 4).

direction or color. This allows us to fit models that are identical to those used in the unimanual task (**Figure 2**). In the bimanual task, the final RTs (RT$^{2nd}$) are well described by the fits to the unimanual double-decision RTs (**Figure 5**). We illustrate this in two ways. In the figure, the solid traces are not fits to the bimanual data; they are fits to the unimanual data shown in **Figure 2B**. Clearly the choice probabilities and RTs in the bimanual task are well captured by the model fit to the unimanual data. The fits to the bimanual data are shown in **Figure 5—figure supplement 1**, and model comparison favors the serial over the parallel model for seven of the eight participants (**Figure 2—figure supplement 1**). Importantly, the participants' behavior was strikingly similar in the unimanual and bimanual versions of the task. The similarity between the two versions of the task is also supported with a model-free analysis. In **Figure 5—figure supplement 2**, we superimpose the accuracy and the RTs for the unimanual and bimanual tasks. There is an almost perfect overlap between these two aspects of choice behavior, providing further support for a common set of processes operating in both versions of the task.

The bimanual task allows us to distinguish between two variants of the serial model that were not distinguishable in the unimanual task. In the first variant, the *single-switch* model, the decision maker only switches from one decision to the next when the first decision is completed. Thus, the decision that terminates first (D$^{1st}$) is the one that is evaluated first, and only then the other decision is evaluated. In the second variant, the *multi-switch* model, the decision maker can alternate between decisions even before finalizing one of them. If little time is wasted when switching, these two models make similar predictions for the RT in the unimanual task: the RT will be the sum of the two decision times plus the non-decision latencies. However, the models make qualitatively different predictions for how the response time for D$^{1st}$ depends on the difficulty of the other decision.

The single-switch model predicts that the RT for D$^{1st}$ is independent of the difficulty of the decision reported second (D$^{2nd}$). This is because D$^{2nd}$ is not evaluated until the first decision is completed. The prediction of the multi-switch model is less straightforward. Suppose that in a given trial the motion decision is easy and the color decision is difficult. If the color was reported first, the motion was probably not evaluated at all before committing to D$^{1st}$, since if it had been evaluated it would most likely have ended before the color decision. In contrast, if both dimensions were difficult, which decision was reported first is largely uninformative about the number of alternations between color and motion that occurred before committing to the first decision; since both decisions take longer to complete, it is possible that both have been evaluated before one of them terminated. Therefore, the multi-switch model predicts that the first decision takes longer the more difficult the other decision is: when D$^{2nd}$ is easy, it is more likely that it was not considered before committing to the D$^{1st}$ decision and thus the average RT$^{1st}$ is shorter.

To disambiguate between the single-switch and multi-switch models, we fit both models to the data from the bimanual task. First, we fit a serial model identical to that of **Figure 2** to the data from the bimanual task. We used the same procedure as in **Figure 2**; that is, we ignore RT$^{1st}$ and fit RT$^{2nd}$ and the choices given to the two decisions. Then, we used three additional parameters to attempt to explain RT$^{1st}$. These parameters are the average time between switches ($\tau_\Delta$), the probability of starting the trial evaluating the motion decision ($p_{m}^{1st}$), and the non-decision time for the first decision ($T_{nd}^{1st}$, see Materials and methods). These parameters only affect RT$^{1st}$; they do not influence RT$^{2nd}$ or the choice probabilities. The three parameters were fit to minimize the mean-squared error between the models' predictions and the data points (**Figure 6**; **Appendix 1—table 2**). The single switch model is a special case of the multi-switch model where $\tau_\Delta$ is very large (i.e. longer than the slowest first decision time).

The model comparison provides clear support for multiple switches. **Figure 6** shows the average RT for the decision reported first (RT$^{1st}$), split by whether the first decision was color or motion, and grouped by either color or motion strength. Both the single- and multi-switch models provide a good explanation of the RT$^{1st}$ when grouped as a function of the coherence of the decision that was reported first (**Figure 6**, panels A and D). However, only the multi-switch model could explain the interaction between RT$^{1st}$ and the coherence of D$^{2nd}$ (**Figure 6B and C**). The graphs show that RT$^{1st}$ is longer when D$^{2nd}$ is more difficult, and this effect was well explained by the multi-switch model. Unlike what is seen in the data, the single-switch model predicts that RT$^{1st}$ should not vary with the coherence of D$^{2nd}$ (flat orange lines in panels B and D). Because we fit the models for each participant individually, we can analyze the frequency of alterations predicted by the model with multiple

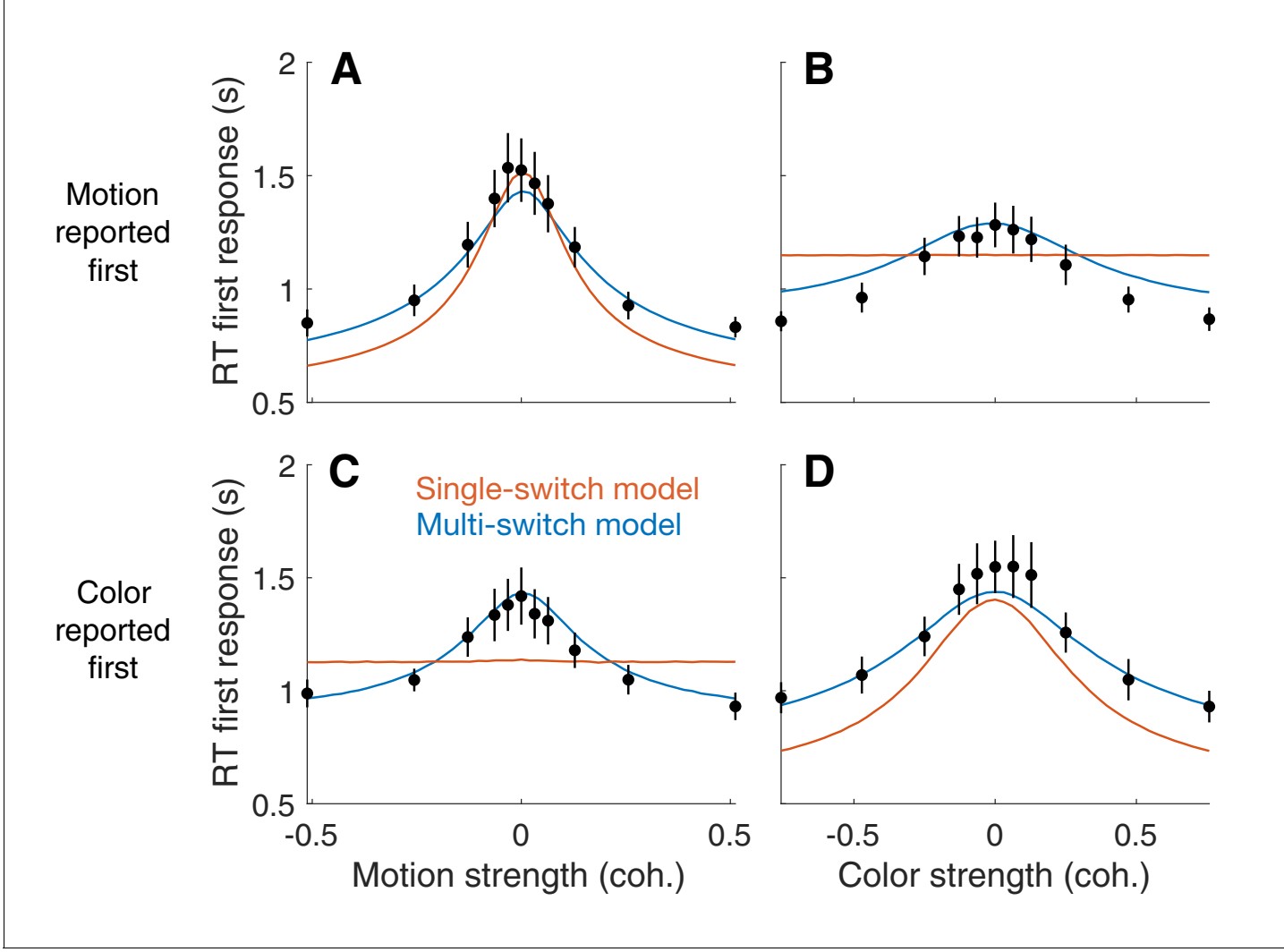

**Figure 6.** First response times in the bimanual task suggest multiple switches in decision updating (Experiment 4). For bimanual double-decisions, participants indicate two RTs per trial. Whereas up to now we have only considered the RT corresponding to completion of both color and motion decisions, the analyses in this figure concern the RT of the first of the two. Symbols are means ± s.e. (N = 8 participants). Curves are fits to single- and multi-switch model (orange and blue, respectively). (A) RT as a function of motion strength when motion was reported first. (B) RT as a function of color strength when motion was reported first. (C) RT as a function of motion strength when color was reported first. (D) RT as a function of color strength when color was reported first. In panels A and D, the first response corresponds to the stimulus dimension represented on the abscissa. The data exhibit the expected pattern of fast RT when the stimulus is strong and slow RT when the stimulus is weak (i.e. near 0). This would occur if the serial processing of motion and color ensued one after the other (single-switch) or with more than one alternation (multi-switch), although the latter provides a better account of the data. In panels B and C, the first response corresponds to the stimulus dimension that is not represented on the abscissa. Here the single-switch model fails to account for the data. If there were only one switch and color terminates first, then the strength of motion is irrelevant, because all processing time was devoted to color. Similarly, if there were only one switch and motion terminates first, then the strength of color is irrelevant, because all processing time was devoted to motion.

switches. For one of the participants, the best-fitting inter-switch interval was greater than the slowest decision time, and thus the model was no different from the single-switch model. For the other seven participants, alternations were sparse: the average inter-switch interval was 704 ± 205 ms (mean ± s.e.m. across participants).

To summarize, the bimanual version of the double-decision task allowed us to infer not only that the two dimensions were addressed serially, but that people may alternate between both attributes of the stimulus in a time-multiplexed manner. The model suggests that alternations were sparse, as if the participants considered one decision for several hundred milliseconds, and switched temporarily to the other decision if they found no conclusive evidence about the first. Moreover, it provides

direct evidence for two termination events, as assumed in our model fits. This rules out a class of models of the double-decision as a race among four accumulations for each of the color-motion combinations, what we term target-wise integration, as these models preclude completion of one decision before the other.

## Experiment 5. Binary-response double-decision reaction time

Up to now, we have observed serial decision making when participants had to provide two answers—that is, four possible responses. A possible concern is that the reason we observed the serial pattern of double-decisions was that it required a quaternary response. We therefore designed a task that involves a double-decision but only a binary choice. Two participants, who had participated in Experiment 4, were asked to report whether the net direction in two patches of random dots were the same or different by pressing one of two response keys with the index fingers of each hand; (*Figure 7A*). The two motion stimuli were presented to the left and right of a central fixation cross *Figure 7A*. The direction (up or down) and strength of motion (three coherence levels) were controlled independently in the two stimuli. Each participant completed four sessions (3072 trials) of this task and additionally completed a single session (768 trials) of a 1D task in which they judged the up/down direction of a single stimulus presented on the left or right side of the screen.

Both participants exhibited accuracy-RT functions that depended on the difficulty of both motion stimuli. *Figure 7B* shows the proportion of correct choices plotted as a function of the coherences for both the 1D (up-down) and 2D (same-difference) trials. The RTs associated with same-different judgment were almost twice as long as the RTs from a 1D direction judgment. Part of this difference might be attributed to the conversion from two direction judgments to the same-different response, but that should not depend on difficulty and it is hard to reconcile this with the magnitude of the difference. Instead they suggest additive decision times. The horizontal red and blue lines in *Figure 7B* are fits to a drift diffusion model that assume the 2D same/different decision is formed from two 1D direction decisions under serial and parallel models, respectively. We constrained the fits in the 1D and 2D tasks to share the same sensitivity to motion strength and the same up-down bias (see Materials and methods, *Equation 2* and *Appendix 1—table 3*). Bayes factor for both participants favored the serial model ($\log_{10} \mathrm{BF} = 49.6$ and 31.6 for S7 and S12, respectively).

The comparison of serial vs. parallel rests on an understanding of the way distributions of the two up/down decision times are combined to generate the same/different decision times. A possible concern is that the parsimonious drift diffusion models used to estimate these latent distributions is wanting. For example, they assume stationary bounds, which distorts the shapes of the distributions (*Drugowitsch and Moreno-Bote, 2014*). We therefore conducted the model comparison using an empirical method that uses only the observed same/different RTs and tries to account for them solely through combination of latent distributions of up/down decision times (*Figure 7—figure supplement 1*). This analysis also provides strong support for the serial account (*Figure 7—figure supplement 1*; $\log_{10} \mathrm{BF} > 7$ for both participants). Like the color-motion task, there is every reason to assume that the acquisition of evidence from the two patches of random dots occurs in parallel. Yet once again, the pattern of RTs supports serial incorporation into the double decision. The use of a binary response in the same-different task rules out the possibility that the long decision times in our 2D experiments are explained by the doubling of alternatives (Hick's law; *Hick, 1952*; *Luce, 1986*; *Usher et al., 2002*). Moreover, the findings demonstrate that the serial incorporation of evidence into a double-decision is not restricted to different perceptual modalities, such as color and motion.

## Parallel acquisition with serial incorporation model

Taken together, the results from our five experiments suggest that the prolongation of RTs in double-decisions is the result of serial integration of evidence during the decision-making process, independent of the modality of choice implementation and number of response options. Parallel acquisition of the two sensory streams followed by serial incorporation into decision variables reconciles the findings of the short duration experiment (Experiment 2) with those of the double-decision RT experiment (Experiment 1). The variable duration (Experiment 3) and bimanual (Experiment 4) experiments suggest that (i) parallel acquisition and serial incorporation is not limited to the short duration experiment and (ii) serial alternation of color and motion can occur before one process terminates. Here, we develop a conceptual framework that accommodates the findings from all five

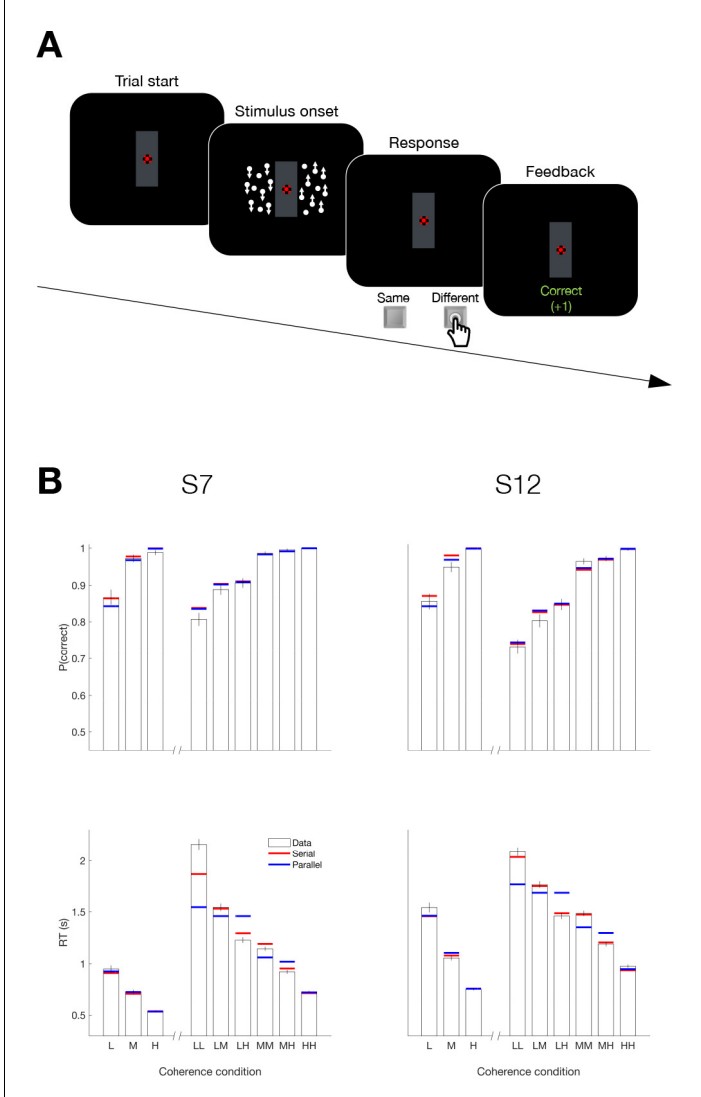

**Figure 7.** Serial decision making in a Same vs. Different task (Experiment 5). (**A**) Task. Two dynamic random dot motion displays were presented in rectangular patches to the left and right of a central fixation cross. The direction and motion strength were randomized from trial to trial and between the patches (up or down × three motion strengths). Participants judged whether the dominant direction of the left and right patches is the same or different and indicated the decision when ready by pressing a response key with their left or right index finger. At the end of each trial, participants received feedback. In a separate block, participants also performed a 1D direction discrimination task in which only one patch of random dots was displayed. (**B**) Results and fits for two participants (columns). *Top*, Proportion of correct choices as a function of the level of motion strength (i.e. unsigned coherence; L = low; M = medium; H = High). *Bottom*, Response times for each level of motion strength. The first three bars represent the direction task where only a single motion stimulus was presented. The six bars on the right of each plot represent the same-different task. Horizontal red and blue lines are fits of serial and parallel drift-diffusion models to the means. Only correct trials were included for RT analyses.

The online version of this article includes the following figure supplement(s) for figure 7:

**Figure supplement 1.** Comparison of parallel and serial rules applied to reaction time distributions in the Same vs Different task (Experiment 5).

experiments with what is known about the neurobiology of similar 1D perceptual decisions. We will proceed by illustrating the steps that underlie the acquisition of evidence samples, their temporary storage in buffers, and their incorporation into the decision variables that govern choice and the two decision times. We first make the case for the buffer using a simulated trial from the short duration experiment. We then elaborate the diagram to account for the serial pattern of decision times when the stimulus duration is longer.

Consider the example in *Figure 8A* of a process leading to a decision in the short duration task. Suppose that visual processing of the 120 ms motion stream gives rise to a single sample of evidence

that captures the information from the brief pulse, and the same is true for the color stream. These samples of evidence are acquired in parallel and placed in buffers, where they can be stored temporarily. The values in these buffers may be thought of as latent instructions to a cortical circuit to update a decision variable ($V_m$ or $V_c$) by some amount ($\Delta V_m$ or $\Delta V_c$). While the samples can be acquired simultaneously, only one sample can update the corresponding decision variable at a time. This is the bottleneck that imposes serial multiplexing in the 2D tasks. One of the samples must be held (buffered) until the other update operation has cleared. If motion is the first to be updated, then $V_c$ cannot be updated until the circuit receiving the motion-update instruction has received it (black arrow). This takes some amount of time, $\tau_{ins}$ (for instruct). The update instruction is realized by an integrator with a time constant ($\tau_v = 40$ ms) leading to slow cortical dynamics (maroon and blue traces).

In this example, each buffer receives all the information available in the stimulus. Were there additional samples in the stimulus, the motion buffer would be ready to receive another sample when it sends its content, whereas the color buffer cannot be updated until it is cleared, $\tau_{ins}$ later. The bottleneck is between the buffer and the update of the decision variable, more specifically, the initiation of the dynamic process that implements this update in a cortical circuit. In this case, there is no consequence beyond a delay, because there is no more evidence from the stimulus after 120 ms.

*Figure 8B* elaborates the diagram in panel A using another trial from the short duration experiment. We now represent the transformation of sensory data to evidentiary samples by applying a stage of signal processing to the raw luminance and color data, $L(x,y,t)$ and $C(x,y,t)$. These functions are just shorthand for the noisy spatiotemporal displays. The motion filter is meant to capture the impulse response of direction selective simple and complex cells in the visual cortex (*Movshon et al., 1978a*; *Movshon et al., 1978b*; *Adelson and Bergen, 1985*; *Britten et al., 1993*; *DeAngelis et al., 1993*), and we assume a similar operation on the stimulus color stream. They are also shorthand for a difference signal, such as right minus left and blue minus yellow. The filtering introduces a delay and a smearing of these streams. While the motion filters must sample the $L(x,y,t)$ at rates sufficient to support the extraction of fast fluctuations and fine spatial displacement, the neurons ultimately pool these signals nonlinearly over space and time (*Britten et al., 1993*; *Zylberberg et al., 2016*). These are the signals represented by the maroon filter traces in *Figure 8B*. This is the convolution of $L(x,y,t)$ and the function in *Figure 3B* (bottom). The same filter is applied to $C(x,y,t)$ to make the filtered color traces (blue). Importantly, for purposes of integrating the information in the color-motion random dot displays, 11 Hz sampling ($\tau_s = 90$ ms) is sufficient. Notice that the filtered representation lasts longer than the stimulus. Therefore, in this case, the decision is based on at least two samples of evidence per sensory stream.

The buffers acquire their first samples at $\tau_s = 90$ ms (*Figure 8B*, arrows ① and ①). The motion buffer is cleared as soon as it is acquired (open maroon rectangle) to instruct a change in $V_m$ (arrow ②). The instruction is received $\tau_{ins} = 90$ ms later. Thus, it is 180 ms after stimulus onset that the neurons representing $V_m$ begin to reflect the motion evidence. We set $\tau_{ins} = 90$ ms mainly to simplify the figure (but see *Figure 8—figure supplement 1*). This unblocks the bottleneck (③), thereby allowing the first color sample to be cleared from its buffer (open blue rectangle) and replaced by a second color sample (filled blue rectangle, ③). Notice that the second motion sample is also acquired at $t = 180$ ms, that is, $\tau_s$ after the first acquisition (and its immediate clearance). The first color sample instructs $V_c$, thus blocking other updates for $\tau_{ins}$ (④) and is first registered by $V_c$ at $t = 270$ ms, which unblocks the bottleneck (⑤). Because we are assuming alternation in this example, this leads to the second update of $V_m$ (⑥). With the motion buffer available, it would be possible to obtain a third sample from the motion stream at $t = 270$ ms, but the filtered signal has decayed to nearly zero, and we assume extinction of the stimulus is registered by the brain in time to terminate sampling. Upon receipt of $\Delta V_m$, the bottleneck is unblocked ($t = 360$ ms; ⑦) and the second color sample is cleared from its buffer ($t = 360$ ms) to instruct $V_c$ (⑧). There is no signal left to integrate, and the decision is made based on the signs of $V_m$ and $V_c$. Thus, the decision is based on simultaneous (parallel) acquisition of two samples of evidence, which are incorporated serially into their respective decision variables.

The exercise helps us appreciate how a stream of evidence lasting only 120 ms could lead to a double-decision 400–600 ms later (*Figure 3A*). It also illustrates the compatibility of parallel acquisition and serial incorporation into the decisions, and it suggests that serial processing is imposed at the step between buffered samples and incorporation into the decision variables. This is the 'response

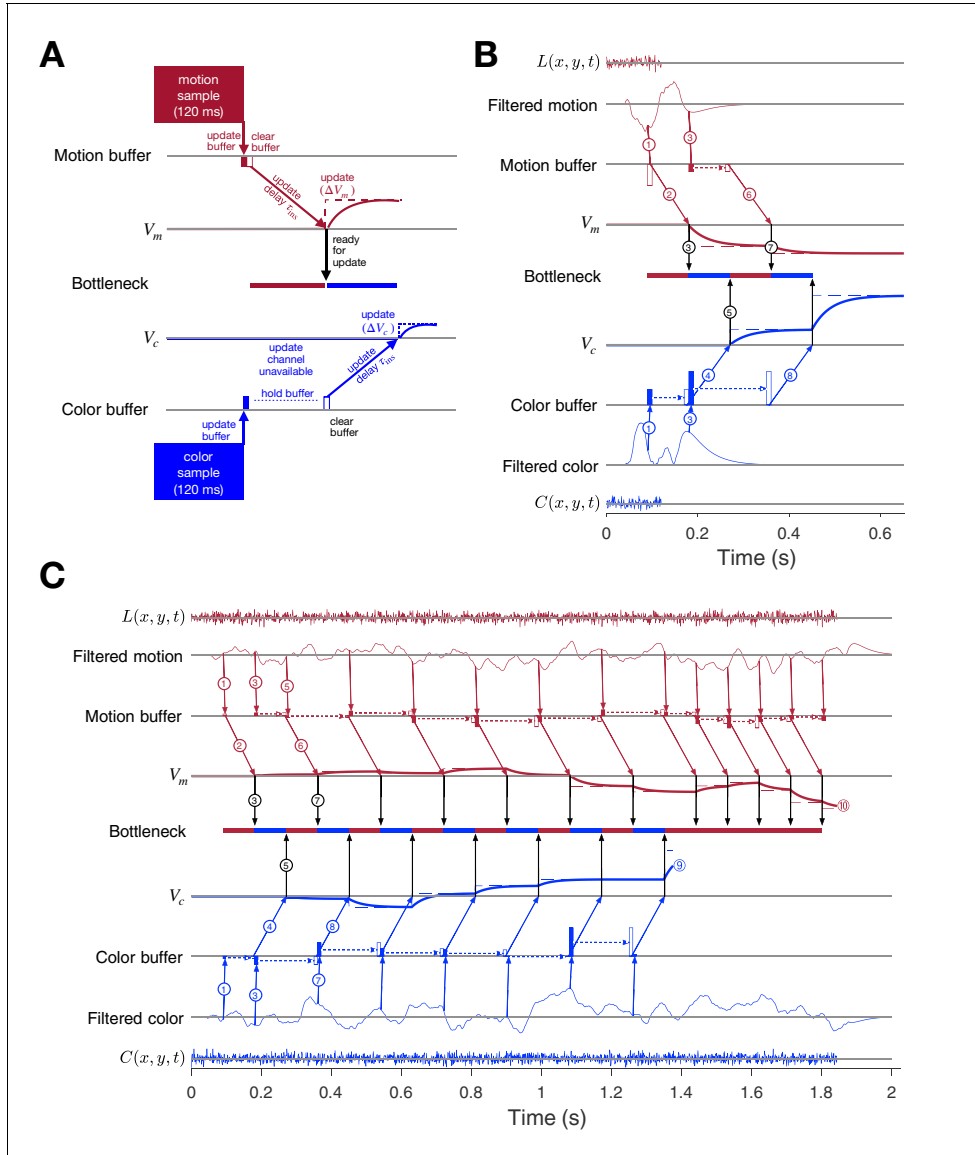

**Figure 8.** Parallel acquisition of evidence and serial updating of two decision variables. An elaborated drift diffusion model permits reconciliation of the serial processing implied by the double-decision choice-RT experiment and the failure to observe interference in choice accuracy when the color-motion stimulus is restricted to a brief pulse. The main components of the model are introduced in panel A and elaborated in panels B and C. In all panels, maroon and blue indicate motion and color processes, respectively. (**A**) Simulated trial from the short duration experiment (Experiment 2). Information flows from top to middle graphs for motion; and from bottom to middle graphs for color. Time is left to right. The evidence from both color and motion is extracted from the 120 ms random dot stimulus in parallel. Both can be stored temporarily in separate buffers (filled rectangles), which send an instruction to the circuits representing the respective decision variables in their persistent firing rates. The instruction is to change the firing rate by an amount ($\Delta V_m$ or $\Delta V_c$). This latency from clearance of the sample from the buffer to receipt of the $\Delta V$ instruction takes time ($\tau_{ins}$, diagonal arrows), and this is followed by the realization of the instruction in the evolving firing rates of cortical neurons (smooth colored curves). In the example, the $V_m$ is the first to update. A central bottleneck precludes updating $V_c$. The bottleneck is unblocked when the $\Delta V_m$ instruction is received by the circuit that represents the motion decision variable (black arrow). This allows the buffered evidence for color to update $V_c$. Open rectangle represents clearance of the buffer content, which occurs immediately for motion and after a delay for color in this example. Dashed lines associated with the decision stage show the instructed change in the decision variable ($\Delta V_m$ and $\Delta V_c$). Smooth colored curves show the evolution of the decision variables. (**B**) Elaboration of the example in panel-A. The boxes representing the 120 ms stimulus are replaced by the two outer rows: (i) raw luminance and color data stream, $L(x,y,t)$ and $C(x,y,t)$, respectively, represented as biased Wiener processes (duration 120 ms); (ii) filtered evidence streams containing the relevant motion (right minus left) and color (blue minus yellow) signals. The filters introduce a delay and smoothing. The filtered signals can be sampled by the buffer every $\tau_s$ ms, so long as the buffer is available (i.e. empty). The bottleneck shows the process that is accessing the update channel. Other than the first sample, the prioritization is equal and alternating. Only one process can update at a time. Circled numbers identify the key events described in Results. Events sharing the same number are approximately coincidental. (**C**) Example of a double-decision in the choice-RT task.

*Figure 8 continued on next page*

*Figure 8 continued*

The first eight steps parallel the logic of the process shown in panel B. The decision variables then continue to update serially, in alternation, until $V_c$ reaches a terminating bound ⑨. The decisions then continues as a single-dimension motion process until $V_m$ reaches a terminating bound (⑩). Note that the sampling rate is the same as it was in the parallel phase, whereas during alternation it was half this rate for each dimension. Bound height is indicated by ⑨ and ⑩.

The online version of this article includes the following figure supplement(s) for figure 8:

**Figure supplement 1.** Example of a 1D choice-reaction time trial.

**Figure supplement 2.** Example of a 2D decision with one switch after parallel acquisition.

**Figure supplement 3.** Example of a 2D decision with stochastic switching.

selection' bottleneck hypothesized by *Pashler, 1994* and others (e.g. *Marti et al., 2012*; see Discussion).

The idea extends naturally to double-decisions that are extended in time. *Figure 8C* illustrates a simulated double-decision in a free response task. The double-decision is made once both decision variables reach their terminating bounds. The example follows the same initial steps as the short duration experiment, except that when the second motion and color samples are cleared from their respective buffers, they are replaced with a third sample. Notice that beginning with the third motion sample, the interval to the next sample has doubled (180 ms), because the example posits regular alternation (for purposes of illustration only; see *Figure 8—figure supplement 2* and *Figure 8—figure supplement 3*). This longer interval begins with the second sample. From that point forward, until the color decision terminates, the streams are effectively undersampled. Decision processes ignore approximately half of the evidence supplied by the stimulus. This is because both streams supply independent samples of evidence at a rate greater than 5.5 Hz (i.e. an interval of 180 ms).

In the example, it is $V_c$ that reaches the bound first ($T_c \approx 1.4$ s; ⑨). There may be no overt behavior associated with this terminating event, as in the eye and unimanual reaching tasks, but direct evidence for this termination is adduced from the bimanual reaching task. From this point forward, the processing is devoted solely to motion until it terminates at a negative value of $V_m$ (⑩). Notice that when the bottleneck is unblocked, there is always a buffered sample ready to be cleared, and this occurs at intervals of $\tau_{ins} = \tau_s = 90$ ms. The process is now as efficient as a single decision process. Indeed, a simple 1D decision about motion (or color) is likely to involve the same instruction delays and bottleneck (*Figure 8—figure supplement 1*). If $\tau_s = \tau_{ins}$, then like the first sample of motion, all subsequent samples of motion could pass immediately from the buffer to update $V_m$ without loss of information. The model is thus a variant of standard symmetrically bounded random walk or drift-diffusion (*Laming, 1968*; *Link, 1975*; *Ratcliff, 1978*; *Shadlen et al., 2006*; *Ratcliff and Rouder, 1998*; *Palmer et al., 2005*). It is compatible with the long time it takes for visual evidence to impact the representation of the decision variable in cortical areas like the FEF and LIP (e.g., ~180 ms).

The diagrams in *Figure 8* are intended for didactic purposes, to lay out the need for a buffer and the seriality imposed by a bottleneck between the buffer and the update of the DV in circuits associated with working memory. The values for the delays and time constants, $\tau_s$, $\tau_{ins}$ and $\tau_v$, were chosen mainly to simplify an already complex diagram, and the same holds for the assumption of strict alternation. The logic does not change if the serial processing were to involve many updates of color or motion before switching to the other dimension (*Figure 8—figure supplement 2* and *Figure 8—figure supplement 3*). The important assumption is that it takes time to update a decision variable, and during this update there is a bottleneck that precludes another update. Importantly, whether alternating, as in *Figure 8C*, or starting one process after completing the other, as in *Figure 8—figure supplement 2*, there is a period of time in which information in the sensory stream is not affecting one of the decisions. This loss is apparent in the additivity of decision times, but it leads to no interference in accuracy in the RT task, because the termination criterion has not changed, and this (and the stimulus strength) determines accuracy. This is the insight that led to the prediction that under certain conditions in which the experimenter controls the duration of the color-motion display, there ought to be interference between color and motion sensitivity (*Figure 4*).

## Discussion

In one sense, the present study extends the framework of bounded evidence accumulation to more complex decisions composed of the conjunction of two decisions about two distinct features. In another more important sense, the findings highlight a bottleneck in information processing that touches on the very speed of thought. The experimental findings demonstrate that a double-decision about the dominant color and direction of motion of a patch of random dots is formed serially. This is surprising, because color and motion are canonical examples of parallel visual pathways from the retina through the visual and extrastriate visual association cortex, and there are compelling demonstrations of this parallel processing on conscious perception (*Cavanagh et al., 1984*; *Cavanagh et al., 1985*; *Carney et al., 1987*). Moreover, the stimulus was designed to minimize interference or competition for spatial attention. It was restricted to a small aperture in the center of the visual field, and the same individual dots supply the motion and color information. It seems fair to say that the deck was stacked in favor of parallel processing. Indeed we confirmed that the color and motion information in the random dot stimulus used here was acquired in parallel.

With one notable exception, there was not a hint of an interaction between color or motion on choice performance in our experiments. That is, changing the difficulty of one dimension, say color, did not affect the perceptual accuracy—or more precisely, sensitivity—to the other dimension, say motion. This held over a wide range of difficulties spanning chance to perfect performance. The one exception was when we controlled viewing duration (*Figure 4*) and this turns out to be explained by a competition of the two streams for processing time, not by an interaction affecting the fidelity of the sensory streams themselves. Had we attended solely to the choice data, we would have likely concluded that the motion and color decisions were formed in parallel, consistent with 40 years of vision science (*Livingstone and Hubel, 1988*; *Ramachandran and Gregory, 1978*).

Evidence for seriality of the decision process is adduced mainly from the pattern of double-decision RTs. The RT is the time from the onset of the color-motion stimulus to the initiation of the movement used to indicate the decision: the sum of the time it takes to complete the double decision, plus time delays that are not affected by task difficulty, termed the non-decision time ($T_{\mathrm{nd}}$). If the color and motion decisions are made in parallel, then the double-decision time is the larger of the two decision times, $\max[T_{\mathrm{m}}, T_{\mathrm{c}}]$. If the decisions are made serially, the double-decision time is the sum, $T_{\mathrm{m}} + T_{\mathrm{c}}$. We focused on the *max* vs. *sum* distinction using a combination of fitting and prediction. The simplest approach relies only on empirical fits of the double-decision RT distributions (*Figure 2—figure supplement 2*) derived from a smaller set of latent distributions of one-dimensional color and motion decision-times, under the appropriate operations for parallel and serial combination (*Equation 6* and convolution, respectively). The approach focuses solely on the RTs and was therefore essential for Experiment 5, where we had no access to the direction choices in the two patches of random dots. It reveals 'decisive' support (*Kass and Raftery, 1995*) for seriality in all but one of the 11 participants (*Figure 2—figure supplement 3*). A drawback of the approach is that it does not constrain the relationship between choice accuracy and decision time. For this, we used a variety of bounded drift-diffusion models. These are the fits shown in *Figure 2*. Here too, we attempted to contrast the *max* and *sum* logic by predicting the RT distribution for the majority of conditions. We fit the choice-RT data from the subset of conditions in which at least one of the stimulus dimensions was at its strongest level. The fits, under the *max* or *sum* rule, supply the marginal distributions of color and motion decision times to predict the RT of the remaining conditions, through application of the same rule. This approach also provides decisive support for the serial model (see *Figure 2—figure supplement 1*).

### The case for buffers and bottlenecks

The strong support for serial processing does not specify where in the processing chain the seriality arises. The answer to this question resolves the apparent contradiction with vision science, and highlights a connection with a body of literature from psychology that addresses the topic of dual task interference, more specifically the psychological refractory period. The key is the short and variable duration experiments (*Figures 3* and *4*). If seriality were imposed at the level of sensory acquisition then when both color and motion are difficult, accuracy on one dimension should come at the expense of accuracy on the other, on average. We did not observe this at short durations, and not for lack of power, as made clear by the interference that was detected at intermediate durations.

Nor did we observe any reduction in accuracy compared to single decisions, and there was no difference in the magnitude and time course over which momentary fluctuations of color and motion predicted the individual choices on single- and double-decisions (*Figure 3B*). These observation also rule out the possibility that there was interference but it was balanced across trials—that is, a mixture of trials in which successful motion processing impaired color processing on half the trials and successful color processing impaired motion processing on the other half.

Thus, the short duration experiment demonstrates parallel processing and the necessity of at least one buffer. The results in the variable duration experiment might lead us to entertain the possibility that only color is buffered, because motion was prioritized. However, the bimanual task demonstrates that motion is not always processed first, and both color and motion are processed before the first process terminates. We therefore conclude that there are two buffers which are capable of holding a sample of evidence about color or motion, respectively, while the other dimension is incorporated into the decision. This places the bottleneck between the buffered evidence and the representation of the decision variable. We believe the bottleneck arises because of an anatomical constraint. It is simply impossible to connect in parallel every possible source of evidence with the neural circuits responsible for representing a proposition or plan. As *Zylberberg et al., 2010* theorized, the brain's routing problem holds the key to why many mental operations operate serially. We will return to this idea after interpreting our results in the context of the neurobiology of decision making. We do this by pursuing the neural correlates of a computational model that supports parallel acquisition of sensory evidence and its serial incorporation into two decisions.

## Connecting computational models to neurobiology

The operations depicted in *Figure 8* are intended to reconcile what is known about the neurobiology of simple 1D decisions with the constraints introduced by the double-decision task. The mathematical instantiation of the model requires only minor modifications of two bounded drift-diffusion processes with temporal multiplexing (see Materials and methods). However, the architecture implied by *Figure 8B,C* facilitates interpretation of the experimental findings in relation to neural processing. In the mathematical depiction of drift-diffusion, the momentary evidence is a biased Wiener process. However, in reality the stimulus is not a Wiener process, nor is the representation of momentary evidence by neurons (*Zylberberg et al., 2016*), which arise through application of a transfer function that effectively spreads the impact of a pair of displaced dots over 100–150 ms (*Figure 3B*, bottom; *Adelson and Bergen, 1985*). Thus, the neural representation of the motion can be approximated by the leaky integral of a biased Wiener process (*Cain et al., 2013*; *Barlow and Tripathy, 1997*). Such smoothing would not be warranted for the detection of fast changes, but it is adequate for a signal that is to be integrated over time. We know less about the filtration of a color difference, but the same logic applies.

The conceptual transition from Wiener processes to discrete samples allows us to appreciate the similarity between the accumulation of evidence from movie-like stimuli and the broader class of decisions based on discrete samples of evidence from the environment and memory. This informs hypotheses about the neurobiology, because the sample of evidence ultimately bears on a decision in units of belief or relative value. That is obvious when considering a choice between items on a menu, but it has been camouflaged to some extent in the perceptual decision-making literature. This is in part because the time-integral of a difference in firing rates from right- and left-preferring neurons is the number of excess spikes for right, which is itself proportional to the accumulated log-likelihood ratio that this excess was observed because motion was in fact rightward (*Gold and Shadlen, 2001*; *Shadlen et al., 2006*; *Beck et al., 2008*). For the wider class of decisions, such difference variables are elusive, whereas the possibility of associating a sample with log-likelihood is a natural dividend of learning and memory (*Yang and Shadlen, 2007*; *Kira et al., 2015*; *Shadlen and Shohamy, 2016*).

The results imply the maintenance of separate decision variables each capable of reconciling decision and choice for the one stimulus dimension. There must be separate control of termination and negligible cross talk. Specifically, the state of the accumulated evidence bearing on the direction of motion does not affect the amount of accumulated evidence required to reach a decision about color dominance, and the same can be said about the state of the accumulated evidence about color on the decision about direction of motion. In the model the decision variables, $V_m$ and $V_c$, represent the integrated evidence for right (and against left) and for blue (and against yellow). Neural correlates of these 1D processes are known, mainly in the parietal and prefrontal cortex (*Gold and*

*Shadlen, 2007*), although they are organized in pairs: $R - L$, $L - R$ (and presumably $B - Y$ and $Y - B$). Each of the four processes is the accumulation of positive and negative increments, and each is terminated by an upper bound. Because evidence for $R$ and $L$ are anticorrelated (likewise for $B$ and $Y$), the pair of opposing processes is approximated by one-dimensional drift-diffusion to symmetric upper and lower terminating bounds. All model-fits adopt this approximation.

An alternative formulation, which we term target-wise integration, would accumulate evidence for the pair of features associated with each choice target (e.g. $RB$, $RY$, $LB$, $LY$). If such mechanism were to terminate when the total accumulation reaches a threshold, it would predict a type of choice-interference such that sensitivity to motion, say, would be impaired when the color strength was high, because the decision time is shortened by the stronger stimulus. We have not pursued all variants of target-wise integration, but critically, the bimanual experiment demonstrates that the double-decision comprises two terminating events.

## Integration as instruction

We find it useful to characterize integration as the implementation of a sequence of instructions to increment and decrement persistent activity in cortical areas that represent the decision variables. In *Figure 8*, the instructed change is realized by simple first order dynamics chosen to approximate neural responses from area LIP. The implementation is merely phenomenological, but it jibes with emerging ideas in theoretical neuroscience that characterize computation as a change in circuit configuration to establish stable states and dynamics (*Remington et al., 2018*). For decision making, it replaces the requirement for continuous integration, with the realization of instructions as if drawn from a memory stage. This characterization also extends to the buffer.

Recall, the buffer was introduced to explain the observation that a brief pulse of color-motion, acquired in parallel, appears to be incorporated into the decision serially. We characterized the length of the buffer—its storage capacity—using the data from the variable duration experiment (*Figure 4*), where we equate it with the duration of parallel acquisition. This is reasonable because thereafter, the process is serial. However, this depiction appears to limit the role of the buffer to the beginning of the decision, and it fails to specify how long the information can be held. If there are alternations between color and motion processing before the first process terminates, as shown in *Figure 6*, then information might be buffered beyond the initial parallel phase. As shown in *Figure 8C*, during alternation a sample might be held for at least $2\tau_{\text{ins}}$—that is, the time it takes the cleared sample to instruct the appropriate decision process and the time the bottleneck is in play while the other dimension performs its update. If the alternations are less frequent, the buffer might need to hold information longer, and if there is only one transition, then the buffer might be expected to hold a sample of information for the duration of the entire first decision (e.g. *Figure 8—figure supplement 2*). There is presumably a limit on how long a sample can be stored, but studies of visual iconic memory suggest that a sample of evidence might be buffered for ~500 ms (*Sperling, 1960*; *Gegenfurtner and Kiper, 1992*).

We conceive of the buffer residing between the cortical areas that represent the filtered evidence and other cortical circuits that represent the decision variables. Notice that the operations depicted in *Figure 8* assign two duties to the buffer: (i) storage of a sample of evidence while the bottleneck precludes updating the associated decision variable and (ii) conversion of the sample into an instruction to update a decision variable by $\Delta V$. These duties could be carried out by different circuits. An appealing candidate for both operations is the striatum. The striatum receives input from the extrastriate visual cortex (*Ding and Gold, 2012b*), and it is known to play a role in connecting value to action selection (*Hikosaka et al., 2014*) as well as working memory (*Akhlaghpour et al., 2016*). In the context of our results, we would characterize the operation as follows. A sample of filtered evidence, represented by the firing rates of neurons in extrastriate cortex (e.g. areas MT/MST) leads to a change in the state of a striatal circuit, such that its reactivation transmits the $\Delta V$ instruction to the cortical areas that represent the decision variables, and this takes time ($\tau_{\text{ins}}$). On this view, the bottleneck is the striato-thalamo-cortical pathway. There has been an observation of the bottleneck in a split-brain patient, supporting such a subcortical bottleneck at least in certain instances (*Pashler et al., 1994*).

A second possibility is that the buffered evidence is stored in visual cortical association areas, especially areas with persistent representations. For example, it has been suggested that short-term visual iconic memory is supported by the slowly decaying spike rates of neurons in area V2 (*O'Herron and von der Heydt, 2009*) and the anterior superior temporal sulcus (STSa)

(*Keysers et al., 2005*). This would place the bottleneck between extrastriate cortex and the parietal and prefrontal areas that represent the decision variable (see also *Marti et al., 2012*). This possibility does not provide an explanation for why communication between these areas would impose a substantial delay (e.g. $\tau_{ins}$).

A third possibility would identify the buffer with control circuitry within the very cortical areas that represent the decision variables. This might seem far-fetched but there is evidence for such an operation in the premotor cortex of mice, where it underlies the implementation of the logical 'exclusive or' (XOR) operation (*Wu et al., 2020*). In that case the bottleneck would be intracortical. It would correspond to the implementation of a circuit state from its 'silent' representation—that is, in cellular and subcellular (e.g. synaptic) states rather than persistent spike activity (*Mongillo et al., 2008*; *Lundqvist et al., 2018*). The bottleneck is the conversion from this state to the establishment of the spiking dynamics that instantiate the $\Delta V$ instruction. This might resemble the recall of an associative memory, which must facilitate the establishment of cortical persistent activity in a state suitable for computation, be it for further updating or comparison to a criterion. The three possibilities are not mutually exclusive; nor are they exhaustive. In any case, the instigating event is the unblocking of the bottleneck, signaled by the circuit that receives the $\Delta V$ instruction.

## The bottleneck

Up to now, we have alluded to the bottleneck as a temporary obstruction to color or motion processing, but the bottleneck itself does not add time. It is the instructive step that takes time ($\tau_{ins}$). This step comprises the conversion of a sample of evidence to a $\Delta V$ instruction and its transmission to a cortical circuit. Indeed, the same delay is encountered in simpler decisions. For example, in the 1D random dot motion task, the incorporation of evidence into the neural representation of the decision variable is first evident 80–100 ms after neurons in area MT exhibit direction selective responses (*de Lafuente et al., 2015*; *Ding and Gold, 2012a*; *Kim and Shadlen, 1999*) and this delay holds for perturbations of the stimulus throughout decision formation (*Huk and Shadlen, 2005*). This is too long to be explained by synaptic latencies. It implies either a complex routing through intermediate structures or more sophisticated processing that serves to facilitate the linkage and/or the conversion of the sample to an instruction suitable for establishing the cortical dynamics that ultimately realize the $\Delta V$ instruction. The delay corresponds to the sum, $\tau_s + \tau_{ins}$ (circles 1 and 2 in *Figure 8*).

Decision variables are represented in the persistent activity of neurons in the parietal and prefrontal cortex of primates. Such persistent activity is associated with working memory, attention and planning. This functional localization conforms to the notion of a 'response selection' bottleneck hypothesized by Harold Pashler to explain dual task interference (*Pashler, 1994*), in particular a phenomenon known as the psychological refractory period (PRP): the prolonged latency of the second of two adjacent decisions without an effect on accuracy. In his and our formulation, it reflects a limitation that restricts the flow of information to affect higher processes such as decision-making and short-term working memory. On initial consideration, there is no obvious reason why the formation of working memory should necessitate a bottleneck. If acquisition can be parallel, why not working memory or the formation of a provisional plan or intention?

Framed in the language of decision-making, seriality arises as a consequence of limited connectivity between the brain's evidence acquisition systems—sensory, memory, and emotion—and the systems that represent information in an intentional frame of reference, that is, as provisional affordances. Any possible intention might be informed by a variety of sources of evidence, which may be acquired in parallel but from different locations in the brain. The brain lacks the anatomy to support independent connections from all sources of evidence to all possible intentions—that is, the circuits that represent them. Instead the communication must share connections, and this invites some form of time-slice multiplexing. It is not possible for every source of evidence to communicate with the circuits that form decisions at the same time. For some dedicated operations, it is likely that many sources of 'evidence' do converge on the same intentional circuitry (e.g. escape response; *Evans et al., 2018*; *Lee et al., 2020*), and the tracts can be established through development. But, for flexible cognitive systems that learn and solve problems, the connections between evidence and intention must be multipotent and malleable, since connecting $N$ sources of evidence and $M$ intentions will need at least $N \times M$ wires if they are connected exhaustively, whereas if they are routed centrally, it will only need $N + M$. This solution necessitates some type of multiplexing (*Zylberberg et al., 2010*; *Feng et al., 2014*; *Musslick et al., 2017*).

## Relation to cognitive tasks

We suspect that the constraints leading to serial processing in the color-motion task also apply to other decisions and cognitive functions. For example, deciding between two familiar food items can take a surprisingly long time when those items are valued similarly. This holds when the items are both highly valued or both undesired or both of moderate value. Like decisions about the direction of random dot motion, there is a lawful relationship between the RT to choose an item and the likelihood that the preference is consistent with one's previously stated value (*Krajbich et al., 2010*; *Krajbich and Rangel, 2011*). Like the choice-RT accompanying 1D motion (or color) decisions, the relationship suggests that some type of process like noisy evidence accumulation—or more generally, sequential sampling with optional stopping—reconciles choice and decision time. However, such expressions of preference differ from perceptual decisions in two important ways. First, there is no objectively correct response, only consistency with the sign of the inequality in the decision-maker's valuations of the individual items, which are ascertained before the experiment. Second, the food items are not shown as a movie and there is no uncertainty about their identity. Therefore, it is not clear what gives rise to independent samples of evidence. *Bakkour et al., 2019* showed that the samples are likely to arise through constructive processes using hippocampal memory systems. This begs the question why this process would unfold in time like a movie of random dots. An attractive idea is that the use of memory guided valuation—in particular the step to enable it to affect a decision variable—encounters a bottleneck. Even if memories could be retrieved in parallel, they would require buffering and serial updates of the decision variable (*Shadlen and Shohamy, 2016*).

While it is unsurprising that a movie of random dots supplies evidence to be incorporated serially toward a decision, it is shocking that two samples of evidence, supplied simultaneously by the same dots and acquired through parallel sensory channels, do not support simultaneous decisions. In the experiments that require prolonged viewing, non-simultaneity manifests in serial time-multiplexed alternation of the decision processes and the failure to incorporate all information in the stimulus stream into one or both decisions. In a free response design, the decision maker compensates by acquiring more evidence, so the interference is not apparent in the accuracy of the perceptual choice. However, if such compensation is precluded by the experimenter (variable duration experiment), the failure to incorporate information can affect accuracy too. That this bottleneck arises despite parallel acquisition of color and motion (or motion from two locations), whether we use one or two effectors to express the decision, and whether we decide between 2 × 2 conjunctions or two categories (same/different) suggests that the bottleneck is pervasive. In addition to the PRP, we suspect that it plays a role in other psychological phenomena, such as post-stimulus masking, the attentional blink (e.g. as shown in the rapid serial visual presentation tasks), and conjunction search (*Potter, 1976*; *Treisman and Gelade, 1980*; *Keysers and Perrett, 2002*; *Marti et al., 2015*; *Marti and Dehaene, 2017*). These phenomena represent forms of sequential interference and all can be stated as challenges to the brain's routing system (*Zylberberg et al., 2010*).

On the other hand, one must wonder if the brain can ever take advantage of parallel acquisition to perform cognitive functions in parallel. It certainly seems so to a musician using their feet and hands to convey time and sonority on a piano or polyrhythm on a drum kit. Yet the time scales of alternation discussed in this paper are on the order of 10 Hz. It seems possible that we achieve parallel processing despite the bottleneck by enhancing signal-processing at the filter stage before the bottleneck and by grouping (or chunking) processes after the bottleneck in higher order controllers of movement and strategy. For example, face selective neurons compute conjunctions of features in less than 100 ms (*Freiwald and Tsao, 2010*). This is just one example of the sophisticated properties of association sensory neurons in the extrastriate visual cortex, and analogous operations are presumed to occur in secondary somatosensory cortex and belt regions of the auditory cortex (e.g. *Bizley and Cohen, 2013*; *Moses et al., 2016*; *Jiang et al., 1997*). Similarly, complex movement sequences and the rules to coordinate them may be specified in premotor cortex or at the level of the controller. If so, then the only way to overcome the bottleneck is to develop the expertise of the reader or the musician/athlete, leaving most of flexible cognition to negotiate the bottleneck between the acquisition of information and its incorporation into representations that support states of knowledge: decisions, working memory, plans of action. It is the price the brain pays to use its senses (and memory) to bear on a plethora of possible intentions, despite its limited connectivity. The payment is in time, but in another sense, it is

time well spent, for without seriality of thought there is no contour to our experiences, no appreciation of cause and consequence, no meaning or narrative.

## Materials and methods

### Participants

Thirteen participants (five male and eight female, age 23–40, median = 26, IQR = 25–32) provided written informed consent and took part in the study. All participants had normal or corrected-to-normal vision and were naïve about the hypotheses of the experiment. The study was approved by the local ethics committee (Institutional Review Board of Columbia University Medical Center).

### Apparatus

Visual stimuli were displayed on high resolution CRT monitors. The experiments were conducted in two labs. *Table 1* lists the display parameters used in all experiments. In the eye-tracking experiments, a head- and chin-rest was used, and eye position was monitored at 1 kHz using an Eyelink 1000 device (SR Research Ltd., Mississauga, Ontario, Canada). In the reaching task participants used robotic handles (vBots, *Howard et al., 2009*) to indicate their choices, and movement trajectories were recorded at 1 kHz. The experiments were run using Matlab and Psychtoolbox (*Brainard, 1997*) and for the online experiments jsPsych (*de Leeuw, 2015*).

**Table 1.** Experimental parameters.
Experiment 1. Double-decision reaction time (eye and unimanual), Experiment 2. Brief stimulus presentation (eye), Experiment 3. Variable-duration stimulus presentation (eye), Experiment 4. Two-effector double-decision reaction time (bimanual) and Experiment 5. Binary-response double-decision reaction time.

| | Exp 1.-eye | Exp 2. | Exp 3. | Exp 1.-uni. & Exp 4. | Exp 5. |
|---|---|---|---|---|---|
| Dot density (dots deg$^{-2}$ s$^{-1}$) | 15.3 | 15.3 | 16 | 16 | 16 |
| Dot speed (deg/s) | 1.67 | 1.67 | 5 | 5 | 5 |
| Dot diameter (deg) | 0.075 | 0.075 | 0.061 | 0.082 | S7: 0.098; S12: 0.119 |
| Fixation marker diameter (deg) | 0.4 gray circle | 0.4 gray circle | 0.6 red cross and bullseye | 0.6 red cross and bullseye | 0.6 red cross and bullseye |
| Random delay (s) | 0.1–0.5 | 0.1–0.5 | 0.5–0.8 | 0.5–0.8 | 0.4–0.8 |
| Visual target diameter (deg) | 0.4 | 0.4 | 1.2 | N/A | N/A |
| Target eccentricity (deg) | 6 | 6 | 15 | N/A | N/A |
| Movement initiation | gaze > 2.5˚ | gaze > 2.5˚ | gaze > 3˚ | hand > 1 cm | key press |
| Target detection window (radius) | 3˚ | 3˚ | 2.4˚ | 0.75 cm | N/A |
| CRT | Vision Master 1451 | Vision Master 1451 | Sony CRT CPD-G420S | Dell CRT P1110 | N/A |
| Refresh rate (Hz) | 75 | 75 | 75 | 75 | 60 |
| Resolution (pixels) | 1400 × 1050 | 1400 × 1050 | 1280 × 1024 | 1280 × 1024 | S7: 1280 × 720; S12: 1440 × 900 |
| Pixels per degree | 39.6 | 39.6 | 32.7 | 24.3 | S7: 40.94; S12: 33.45 |
| Viewing distance (cm) | 55 | 55 | 50 | 38 | S7: 54; S12: 38 |
| Blue cd/m$^2$ [M(SD)] | N/A | N/A | 25.20 (0.81) | 12.16 (1.93) | N/A |
| Blue CIE x/y [M] | N/A | N/A | x = 0.26, y = 0.24 | x = 0.27, y = 0.24 | N/A |
| Yellow cd/m$^2$ [M(SD)] | N/A | N/A | 22.98 (0.05) | 12.68 (1.80) | N/A |
| Yellow CIE x/y [M] | N/A | N/A | x = 0.54, y = 0.38 | x = 0.54, y = 0.38 | N/A |

## Overview of experimental tasks

Participants sat in a semi-dark booth in front of a CRT monitor. They were required to decide the net direction and the dominant color in a patch of dynamic random dots. Individual dots were displayed for a single video frame (1/75 s). Task difficulty for motion was conferred by the probability that in frame $n + 3$ (i.e. $\Delta t = 40$ ms), it would be displaced in apparent motion vs. randomly replaced in the aperture. We prepend the probability by plus or minus to indicate the direction, and refer to this signed quantity in units of coherence (coh). For color, task difficulty was conferred by the probability that a dot would be colored blue or yellow on each frame. We refer to the signed quantity, $2(p_{\text{blue}} - 0.5)$, as the color coherence. Both coherences share the range $\{-1, 1\}$. Throughout, we use positive coherence for rightward and blue dominant stimuli. The coherences were stationary during a trial but randomized independently across trials. A calibration procedure was used to match the luminance of the blue and yellow for each participant (see below). For the first experiment (choice-RT, participants S1– 3), the color of the dots in the first three frames of a trial was balanced to give no net color information. The procedure was intended to match the state of the motion stimulus which is effectively zero-coherence until the fourth video frame. Subsequent experience demonstrated that this procedure was unnecessary, and we discontinued this practice for the other experiments.

We conducted three types of tasks: (*i*) double-decision (2D), in which both the dominant color and motion direction were reported on each trial (color-motion task, Experiments 1–4); (*ii*) single-decision (1D), in which only the dominant color or net motion direction were reported, as in *Mante et al., 2013* (motion-only and color-only tasks, Experiments 1–3; motion-only, Experiment 5); and (*iii*) same vs. different, in which the directions of motion in two patches of random dots are compared (Experiment 5). For the color-motion task, four choice targets appeared at four corners of the display, evenly spaced from each other and the same distance from the fixation spot. The top two targets were colored yellow and the bottom two blue, consistent with the color choices they indicate. For example, to report rightward motion and yellow color, the participant would saccade (Experiment 1 [eye], 2. and 3) or reach (Experiments 1 [hand] and 4) to the top right target, which was yellow. For the motion-only task, two white targets were shown to the left and right of the stimulus. For the color-only task, one blue and one yellow target were presented above and below the center of the screen. For the 1D task, the signed coherence of the 'irrelevant' dimension was varied from trial to trial just as in the 2D task. For the same vs. different task, participants reported by pressing one of two keys.

Note that in general, the 1D experiments do not provide a principled comparison to the 2D decision tasks (*Pashler, 1999*; *McLeod, 1977*). The error rate on difficult conditions are different (0.5 and 0.75 for 1D and 2D, respectively), and therefore there is no reason to expect a decision maker to implement the same speed-accuracy settings in the two contexts (*Churchland et al., 2008*; *Usher et al., 2002*; *Ditterich, 2010*). This consideration applies to the RT experiments, naturally, but also to experiments in which the duration is controlled experimentally (e.g. Experiment 3), where it is known that decision makers also control termination criteria (*Kiani et al., 2008*; *Kang et al., 2017*). This consideration does not apply to the experiment using very brief stimuli (Experiment 2) because differences in termination criteria would not be expected to play a role; all difficult trials would benefit from more information. Therefore, except for Experiment 2, we do not dwell on the results of the 1D experiments in this paper (but see *Figure 2—figure supplement 6* and *Appendix 1—table 4*). The data are also included in the online materials.

For each experiment, the sample size was determined based on prior psychophysics studies with within-subject designs (*Palmer et al., 2005*; *Resulaj et al., 2009*; *Zylberberg et al., 2012*; *Kiani et al., 2014*). We recruited three participants for the first and second experiment (Choice-reaction time task and short duration, eye). For the remaining experiments we recruited 2–8 participants. A larger number was necessary for the arm experiments because fewer trials per hour are acquired and the effort is greater. Unless otherwise stated, participants were randomly allocated to experiments.

## Experiment 1. Double-decision reaction time (eye and unimanual)

Two versions of this experiment were conducted, one where participants indicated their decision with an eye movement (Exp. 1-eye) and one where they indicated their decision with a unimanual reach (Exp. 1-unimanual). Here, we describe the Methods for the eye movement task. The methods

for the unimanual task are described with Experiment 4 - Two-effector double-decision reaction time (bimanual).

Three participants (1 male and 2 female, aged 25–40) performed the task in which they could view the random dots until ready with a response (*Figure 1a*). Participants were required to fixate a central spot for 0.5 s to initiate a trial. After a random delay, a patch of dynamic random dots appeared which were restricted to an invisible circular aperture (5° diameter), centered on the fixation spot. The random dots were extinguished when the participant initiated the choice response. Participants were required to respond within 5 s of the stimulus onset. Trials in which no response was initiated and those aborted by breaking fixation were repeated at a later time in the experiment. When the participant indicated the decision, a gray circle was drawn around the correct choice. The participant earned one point if both decisions were correct; and otherwise they lost one point. The two outcomes were signaled by different sounds. Participants were instructed to maximize points earned per time by responding as fast and accurately as possible. To encourage this, the points accumulated in the current block, as well as a graph of their scores in each 1 min period in the block, were displayed as feedback.

Participants performed three trial types—1D color-only, 1D motion-only and 2D color-motion decisions—in 24–49 interleaved blocks (~13 min, 200 trials) over 11–17 days. Five levels of difficulty (i.e. nine signed coherences including 0) for color and motion were employed for all trial types, although for the 1D task, two of the strengths were not used for the irrelevant dimension. The set of non-zero motion strengths was doubled for one participant (S1) because they failed to achieve >90% correct at coherence = 0.256 during training. Likewise, the range of color strengths was doubled for two participants (S1 and S3). We therefore use a normalized scale for plotting the data in *Figure 2A*. The actual coherence values (*Table 2*) are used in all fits.

The instructions for the 2D task encouraged participants to make a combined judgment, rather than judging color and motion serially (verbal instruction provided by experimenter: "Your task is to answer based on both motion and color. When the motion is right and color is yellow, the answer is top right, and when the motion is right and color is blue, the answer is bottom right."). Participant S1's performance on the motion was at chance level in the last three blocks (one motion-only and two color-motion decision blocks; accuracy = 38–50% within each of the three blocks, containing 287 out of 4911 trials performed by S1). Those blocks were excluded from all analyses (minimum accuracy [motion or color] of all other blocks of S1 and every block of other participants: 68%), leaving 4624–10,969 trials across the participants (2491–5968 1D trials and 2133–5001 2D trials).

### Training sessions

Participants completed 11–13 training blocks (13 min) over 4–7 days, beginning with either an easy motion or color 1D task. S1 and S2 began training with the 1D color-only task; S3 began training with 1D motion-only. All were then trained on the other 1D task, before they were trained on the 2D task. For initial training, viewing duration was controlled by the experimenter. The incorporation of

**Table 2.** Motion and color strength parameters.
For 2D trials all combinations of motion and color strengths were used. For 1D trials all strengths were used for the dimension that informed the decision but some strengths (*) were omitted for the other dimension.

| Experiment | Participant | Motion strengths | Color strengths |
|---|---|---|---|
| 1. Double-decision RT (eye) | S1 | 0, 0.064*, 0.128, 0.256*, 0.512 | 0, 0.062*, 0.124, 0.245*, 0.462 |
| | S2 | 0, 0.032*, 0.064, 0.128*, 0.256 | 0, 0.031*, 0.062, 0.124*, 0.245 |
| | S3 | 0, 0.032*, 0.064, 0.128*, 0.256 | 0, 0.062*, 0.124, 0.245*, 0.462 |
| 2. Brief stimulus presentation (eye) | S1–3 | 0, 0.064*, 0.128, 0.256*, 0.512 | 0, 0.124*, 0.245, 0.462*, 0.762 |
| 3. Variable-duration stimulus presentation (eye) | S4 | 0.03, 0.063, 0.512 | 0.052, 0.104, 0.758 |
| | S5 | 0.044, 0.084, 0.512 | 0.046, 0.104, 0.758 |
| 4. Two-effector double-decision reaction time (bimanual, same strengths for unimanual) | S6–13 | 0, 0.032, 0.064, 0.128, 0.256, 0.512 | 0, 0.064, 0.128, 0.250, 0.472, 0.758 |
| 5. Binary-response double-decision RT | S7 and S12 | 0.128, 0.256, 0.512 | N/A |

weaker stimulus strengths and the range of stimulus durations were adjusted progressively. Transitions to the next level were made if the participant met fixation requirements and achieved >90% accuracy on the strongest coherence. The aim was to identify four levels of motion strength $\geq 0.032$ and four levels of color strength $\geq 0.031$ in octave steps such that the strongest level (eight times the lowest logit) supported >90% accuracy. We then changed from variable duration to the RT version of the 1D task, again ensuring that the range of difficulties led to at least 90% accuracy for the easiest condition. We then repeated these steps for the other stimulus dimension before introducing the 2D choice-RT task. They received a session of practice to gain familiarity with the 4-choice design. For participants S1 and S3, we made a final adjustment of the difficulty levels. The stimulus strengths were then fixed for all test sessions (*Table 2*).

## Minimum-motion procedure

Prior to the experiment, we calibrated the two colors (yellow and blue) to be equiluminant using the minimum-motion procedure (*Cavanagh et al., 1987*). A flickering vertical sinusoidal grating (spatial frequency 1.25 cyc/deg; temporal frequency 6.25 Hz) is composed as the sum of a blue/yellow counterphase chromatic grating and a light-green/dark-green counterphase luminance grating. The spatial and temporal phases of the gratings differ by $\pi/2$ radians, like the functions $\sin(x)\sin(t)$ and $\cos(x)\cos(t)$. If yellow is more luminant than blue, the grating appears to move in one direction (e.g. left), and vice versa. Participants adjusted the luminance of yellow until they did not see motion. Each participants repeated this procedure 24 times, starting from a random luminance value. The mean luminance of yellow was then used throughout the experiment.

## Experiment 2. Brief stimulus presentation (eye)

The same participants from Exp. 1-eye then performed a task that was identical except that the dynamic random dots turned off after 120 ms from the onset. Participants were free to respond after the offset of the dynamic random dots. The RT in this task was measured as the time between the onset of the stimulus and the response (the time the gaze left the center of the screen). Participants completed 35–43 test blocks (~13 min) over 12–19 days, comprising 7309–7745 trials (4056–4192 1D trials and 3240–3466 2D trials). The stimulus strengths are in *Table 2*.

## Experiment 3. Variable-duration stimulus presentation (eye)

Two participants (female, aged 26 and 32; both right-handed) completed 12–26 sessions (after the training sessions), each requiring 1–2 hr. The task alternated between blocks of 72–144 trials where participants either performed the 2D variable duration task, a 1D variable duration task or a 2D choice-RT time task. The majority of blocks were 2D variable duration (~11,800 trials per participant). Ten stimulus durations, ranging from 120 to 1200 ms (in steps of 120 ms), were presented in pseudo-random order. Warning messages were displayed if participants initiated an eye movement before the end of the stimulus ('too early!') or if a movement was not initiated within five seconds of stimulus offset ('too slow!'). In both cases, the trial was aborted and repeated at a later, randomly determined, point within the same block. Participants received auditory and visual feedback for correct and error trials, as in Experiment 1. They were instructed to be as accurate as possible and received feedback about the percentage of correct choices for each decision dimension at the end of each block.

Only three levels of difficulty were used for each dimension: one easy and two difficult coherence levels. The easy coherence level was 0.512 for motion and 0.758 for color. The two difficult coherence levels were adjusted individually in order to match color and motion performance. Specifically, low coherences (hard) for each dimension were chosen to yield 65% and 80% accuracy on each dimension, respectively, based on participants' performance in the final two training sessions (2D choice-RT). All 6 × 6 combinations of signed motion × signed color were presented (see *Table 2*). Since the main model predictions are based on a comparison of trials with hard-hard vs. hard-easy combinations, easy-easy combinations were only presented in ~2.4% of trials. All other coherence combinations were presented with equal frequency and counter-balanced within each stimulus duration. Participants also completed 2160 trials each of motion-only and color-only trials and 1296 trials of the 2D choice-RT task which were included to ensure that they maintained appropriate speed-accuracy trade-offs throughout the experiment.

## Training sessions

Participants first completed 6–9 training sessions. In the first two sessions, they were trained on a variable duration task. Stimulus durations were drawn randomly from a truncated exponential distribution ranging between 500 and 2000 ms (session 1) or 100 and 1600 ms (session 2). Participants first completed 1D-motion and 1D-color tasks in separate blocks (order counterbalanced across participants), followed by the 2D task. The following instructions for the 2D task were presented on the screen: "Your task is to judge both the direction and color of the dots by moving your eyes to the corresponding left/right, yellow/blue target'. In the remaining training sessions, participants mainly performed a 2D RT task until they reached stable performance (at least 60% accuracy on the second coherence level for both decision dimensions, with little change in choice performance or RTs over blocks). Occasionally, additional 1D blocks were introduced to attain similar performance levels for motion and color judgments. Throughout training, all six coherence levels for motion $\{0, \pm0.032, \pm0.064, \pm0.128, \pm0.256, \pm0.512\}$ and color $\{0, \pm0.064, \pm0.128, \pm0.250, \pm0.472, \pm0.758\}$, and all their possible pairwise combinations, were presented. For double decisions, this makes 121 unique signed-coherence combinations.

## Isoluminance calibration

At the start of the experiment, participants completed a flicker fusion procedure to match luminance of yellow and blue. A square ($4.9° \times 4.9°$) was presented in the center of the screen. The color of the square flickered at 37.5 Hz between blue and yellow. For efficiency, we only explored values $[R\,G\,B] = [0\,x\,x]$ and $[R\,G\,B] = [y\,y\,0]$, for blue and yellow, respectively, where $x, y \in \{\mathbb{N} : 0, \cdots, 255\}$. Participants pressed the left or right arrow key to minimize the perceived flicker. One key changed $x$ and $y$ by +1 and $-1$, respectively, and the other key had the opposite effect. Participants pressed the space bar to signal the subjective point of minimal flicker. This procedure was repeated 10 times, each time starting with new initial values $[x\,y]$, chosen pseudo-randomly, such that either yellow or blue began darker (counter-balanced across trials). The precise initial values were equidistant from 225: between 195 and 200 for the darker color and 250 and 255 for the brighter color (e.g. blue: [0 197 197]; yellow: [253 253 0]). This ensured sufficient contrast to induce the perception of flicker at the start of each trial. The averages across the 10 trials were adopted as the isoluminant setting for the participant. After the procedure, participants were presented with a single trial with the obtained color values and were asked to report if they perceived flicker. If they did, the procedure would be repeated; but this never occured. The same calibration procedure was also used for Experiment 4.

## Experiment 4. Two-effector double-decision reaction time (bimanual)

This experiment examined a double-decision task in which the report was made with a single effector (unimanual) or with two effectors (bimanual). We present the unimanual results with Experiment 1, although we describe the methods here.

Twelve right-handed participants were initially recruited for the experiment. After training, eight participants were selected for the actual experimental sessions based on their overall performance. Participants completed two test sessions with a unimanual version of the task and two test sessions with a bimanual version (order counterbalanced across participants). In each experimental session, all 121 color-motion combinations (see *Table 2*) were presented pseudo-randomly in 12 blocks of 96 trials each (total of 1152 trials per participant).

Unlike the eye experiments, no choice targets were present on the screen. Instead, there were arrow icons that indicated the mapping of color and motion to forward/backward (appropriately colored) and left/right directions of the hand (*Figure 5A*). The mapping of blue/yellow to bottom/top target locations was counterbalanced across participants. The movements themselves were restricted to virtual channels in the plane. In the unimanual task, participants moved a single robotic handle with either their left or right hand (counterbalanced across each half of a session) in one of the four diagonal target directions (2 color × 2 motion; as in the other experiments). In the bimanual task, participants used two separate robotic handles to move their left and right hands in a left/right (motion judgments) and forward/backward (color judgments) direction, respectively (hand assignments counterbalanced across participants). Feedback about the hand position(s) was provided by two black bars on top of the arrow icons (for clarity shown as gray in *Figure 5A*). Participants were instructed to move each bar in the chosen direction until their hand(s) reached a virtual 'wall' at the end of the channel, at which

point their decisions were registered. Movement distances between starting positions and target locations were identical in the uni- and bimanual task (5 cm). On 2D trials, the random dots were extinguished when both decisions were indicated, that is when the hand left the home position in the unimanual task and when both hands had left the home position in the bimanual task. Warning messages were presented if participants initiated a response before stimulus onset or within 200 ms of stimulus onset ('too early') or when RTs exceeded 5 s ('too slow!'). In both cases, the trial was aborted and was repeated at a later, randomly determined, trial within the same block.

Once participants indicated their decision, a green or red frame appeared around each response arrow to indicate whether the choice on the corresponding dimension was correct. If both decisions were correct, additional auditory feedback was provided (700 Hz tone) and the participant earned one point. Participants were instructed to maximize points by responding as fast and accurately as possible. At the end of each trial, they received feedback regarding their current rate of rewards (points/min) as well as a graph of their scores in each 2 min period over the last 10 min. To further motivate participants to adopt appropriate speed-accuracy trade-offs, the feedback duration was longer for errors than correct responses, thus delaying the onset of the next trial (correct: 1.25 s; error on one dimension: 2 s; error on both dimensions: 3 s). At the end of the trial the robotic interface actively moved the hand(s) back to the home position(s).

### Training

All participants completed three to four initial training sessions, using the version of the task that they were assigned to first (uni- or bimanual, counterbalanced). In the first two training sessions, participants performed a variable duration task with stimulus durations varying between 500 and 2000 ms. The third training session introduced the choice-RT design. To train participants to maximally separate their two hands in the bimanual version, the RT training task alternated between easy motion and easy color blocks. Participants were encouraged to respond as quickly as possible to the easy dimension while taking more time to make a correct choice on the harder dimension. For participants who were first trained on the unimanual version, stimulus coherences were also presented in blocks of easy motion and easy color to ensure consistency in training across all participants. Participants were invited for the experimental sessions only if their overall rate of warning messages was less than 5% and if their average accuracy was at least 95% on the easy dimension and at least 65% on the 3rd highest coherence level of the harder dimension (e.g. motion: 0.064; color: 0.128).

All participants were immediately trained on the 2D task (without prior 1D training) in order to familiarize them with the motor response required to indicate their double decisions. The following instructions were presented on the screen: 'You will see some dots 'swirling in the wind'. Your task is to judge whether the wind is blowing the dots more strongly to the left or right AND whether the majority of dots is blue or yellow'.

After initial training, participants completed two experimental sessions of the task they had been trained on (either uni- or bimanual RT task). They then completed another practice session, in which they were trained on the other version of the task (either bi- or unimanual RT task), before completing two final experimental sessions with this version of the task. Experimental sessions only differed in motor implementation of decisions (uni- vs. bimanual), but were otherwise identical, and S-R mappings were kept consistent. Participants typically mastered the movements required for both the uni- and bimanual task within the first few blocks of training. All participants included in the final sample were able to report decisions for the high-coherence stimulus component in both the uni- and bimanual task with accuracy >95%.

Note that the data from the unimanual task are presented in *Figure 2B*, (Exp. 1-unimanual). The fits in *Figure 2B*, but not the data, are also displayed in *Figure 5B*.

## Experiment 5. Binary-response double-decision reaction time

The experiment was conducted remotely during the SARS-CoV-2 pandemic (summer 2020). Two participants who had also completed the uni- and bimanual tasks were recruited for this experiment. Participants completed the task online using a Google Chrome browser on Windows 10 and macOS Catalina (version 10.15.4), respectively. Both participants completed eight separate 1 hr sessions within a 2-week time period. The task was programmed in JavaScript and jsPsych (*de Leeuw, 2015*).

During the task, two random dot motion patches with rectangular apertures ($3° \times 5°$) were presented to the left and right of a red fixation cross and separated by a central gray bar ($2° \times 5°$) (*Figure 7A*). Motion direction (up/down) and coherence ($\{0.128, 0.256, 0.512\}$, referred to as low, medium, and high) of the two stimuli were independent of each other. The six unique signed coherence combinations were presented with equal frequency and in randomized order. The stimuli directions and allocation to the left vs. right side of the screen were counterbalanced. Participants had to judge whether the dominant motion directions of the two stimuli were the same or different and indicate their choice by pressing the F or J key with their left or right index finger, respectively, when ready. The response mapping was counterbalanced across the two participants and was shown at the bottom of the screen throughout the task. Visual feedback was provided at the end of each trial. For correct responses, participants won one point. After errors and miss trials (too early/late), participants lost one point. Miss trials were repeated later during the same block. Participants were instructed to try and win as many points as possible and they received an extra bonus of one cent for every point they won. Their point score was shown in the corner of the screen throughout the task and additional feedback about percent accuracy was provided at the end of every block.

Participants first completed three training sessions in which they started on the 2D task without prior 1D training. Participants received the following instructions (shown on screen): 'Your task is to judge whether the dominant motion direction of the dots on the left side is the SAME or DIFFERENT than the dominant motion direction of the dots on the right'. After training, they completed four test sessions of the same-different task (3072 trials). Finally, participants completed a single session (768 trials) of a 1D task in which the random dot motion was restricted to the left or right patch (counterbalanced across trials) and participants had to judge the motion direction (up vs. down) by pressing the M or K key using their right index and middle finger.

At the end of each session, participants completed a separate block of 32 trials with 100% coherence stimuli only (sessions 1–7: same-different task; sessions 8: 1D task). Participants were instructed that decisions in this block would be very easy and that they should respond as fast as they could while still being accurate. The RTs obtained from these blocks (not shown) serve as a check on our estimate of the non-decision time (*Stine et al., 2020*), but they were not used in the analyses. Participants were instructed to maintain fixation throughout the task. At the end of each session, they provided self-report judgments indicating to what extent they kept fixation during the task on a scale from 1 ('not at all') to 4 ('always'). The mean and interquartile range of the reports were 3.75 and 3.5–4 (combined for the two participants). Prior to the experiment, participants completed a virtual chin-rest procedure in order to estimate viewing distance and calibrate the screen pixels per degree (*Li et al., 2020*). This involves first adjusting objects of known size displayed on the screen to match their physical size and then measuring the horizontal distance from fixation to the blind spot on the screen ($\sim 13.5°$).

## Serial and parallel drift diffusion models

Both the serial and parallel models assume that decisions are based on the accumulation of evidence over time. The decision processes for color and motion are described by two independent Wiener processes with drift. The decision variable for one of the dimensions (here motion), evolves according to the sum of a deterministic and a stochastic component:

$$\Delta V_{\mathrm{m}} = \mu_{\mathrm{m}} \Delta t + \mathcal{N}(0, \sqrt{\Delta t}) \tag{1}$$

The deterministic term depends on the drift $\mu_m$,

$$\mu_{\mathrm{m}} = \kappa_{\mathrm{m}}(s_{\mathrm{m}} + s_{\mathrm{m}}^0), \tag{2}$$

where $s_{\mathrm{m}}$ is the stimulus motion strength (signed coherence). By convention, $s_{\mathrm{m}}$ is positive (negative) when the motion is to the right (left). $\kappa_{\mathrm{m}}$ is a parameter that converts coherence to a signal-to-noise ratio, which we fit to the data. $s_{\mathrm{m}}^0$ is a bias that allows us to explain, for example, why left and right responses may not be equiprobable even when there is no net motion in either direction. We model the bias term as an offset in the coherence rather than the starting point of the accumulation. This approximates the optimal way of incorporating a bias in drift-diffusion models when there is uncertainty about the reliability of evidence (e.g. the coherence levels vary across trials) (*Hanks et al., 2011*; *Zylberberg et al., 2018*).

The second term of *Equation 1* describes the stochasticity that affects the evolution of the decision variable. It captures the variability introduced by the stimulus and the brain. This variability is modeled as samples from a normal distribution with zero mean. By convention, the standard deviation is $\sqrt{\Delta t}$, which results in the variance of the decision variable equal to 1 after accumulating evidence for 1 s. This choice does not lead to any loss of generality since for any other value it would be possible to define a new model that has the same behavior in which the variance is 1 and the other parameters are a scaled version of the original ones (*Palmer et al., 2005*). The assumption is restrictive, however, in its requirement that the variance is the same for all motion and color strengths. Based on recordings from direction selective neurons in macaque extrastriate cortex, it seems likely that the variance of the momentary evidence increases with stronger motion, owing to the imbalance between the response of neurons to motion in their preferred vs. antipreferred direction (*Britten et al., 1993*; *Shadlen et al., 2006*). We suspect this leads to an overestimate of the variance at low motion strengths, consistent with underestimation of RT in these conditions (*Figure 2*).

The accumulation process stops and a decision is made when the accumulated evidence reaches one of two bounds. The choice is 'rightward' if the decision terminates at the upper bound, and 'leftward' if it terminates at the lower bound. The decision time is the time $T_{\mathrm{m}}$ that it takes the decision variable to cross the bound. The upper and lower bounds are assumed symmetric with respect to zero. To explain why errors are (often) slower than correct responses, the bounds are allowed to collapse over time. We parameterize the bound as a logistic function with slope $a_{\mathrm{m}}$. The bound reaches a value of $u_{\mathrm{m}}/2$ at $t = d_{\mathrm{m}}$ and approaches 0 as $t \to \infty$:

$$B_{\mathrm{m}}(t) = \frac{u_{\mathrm{m}}}{1 + e^{a_{\mathrm{m}}(t - d_{\mathrm{m}})}} \qquad (3)$$

with lower bound simply $-B_{\mathrm{m}}(t)$. The collapsing bounds explain slower errors because the smaller $|B(t)|$ at later time induces an increased probability that the $V_{\mathrm{m}}$ will terminate in the bound opposite the sign of drift. Although we regard decisions about 0% coherence as neither correct nor incorrect, any bias implies a drift rate $\mu_{\mathrm{m}} \neq 0$ (*Equation 2*). This leads to a small discrepancy between the model fits to the mean RT on these trials (*Figure 2*), because the curves are predictions of the mean double-decision RT for correct choices, but the data plotted at 0% coherence include all trials, just under half of which are effectively slow errors (i.e. choices opposite the bias). This issue affects only the plots of mean RT. All statistical analyses use estimates of the joint probabilities of decision time and choice for each trial.

The same equations describe the decision process for color. We use subscript $c$ instead of $m$ to refer to the color decision, and adopt the convention that positive (negative) evidence supports the blue (yellow) choice. Given a set of parameters ($\Phi_x = [\kappa_x, s_x^0, u_x, a_x, d_x]$), where $x \in \{c, m\}$, we can estimate the probability density function for the decision times $T_x$, and the two possible choices $R$ (right/left for motion and blue/yellow for color). This density function, denoted $p_x^s(T, R)$, depends on the signed stimulus coherence, $s$. We obtain it by numerically solving the Fokker-Planck equation associated with the Wiener process with drift (*Kiani and Shadlen, 2009*), using the numerical method of *Chang and Cooper, 1970*.

So far, the model description applies to single decisions about motion. The serial and parallel models explain how these components are combined to form color-motion double-decisions. In the serial model the accumulation of evidence at any time can only be for color or motion. Therefore, the total decision time $T$ is the sum of the decision times for motion ($T_{\mathrm{m}}$) and color ($T_c$), and the distribution of decision time is given by:

$$p_{\mathrm{serial}}^{s_{\mathrm{m}}, s_c}(T|R_{\mathrm{m}}, R_c) = p^{s_{\mathrm{m}}}(T_{\mathrm{m}}|R_{\mathrm{m}}) * p^{s_c}(T_c|R_c), \qquad (4)$$

where $R_c$ and, $R_{\mathrm{m}}$ are the responses (i.e., choices) for color and motion, respectively, and * denotes convolution.

In the parallel model, both the motion and color are processed simultaneously. Therefore, the decision time is the maximum of either decision time: $\max(T_c, T_{\mathrm{m}})$. We can numerically derive the distribution of decision times from the single-modality distributions by noting that the decision time is equal to $t$ if (*i*) motion ended at time $t$ and color ended before time t, (*ii*) color ended at time $t$ and motion ended before time $t$. Thus,

$$p_{\text{parallel}}^{s_{\mathrm{m}},s_{\mathrm{c}}}(T|R_{\mathrm{m}},R_{\mathrm{c}}) = p^{s_{\mathrm{m}}}(T|R_{\mathrm{m}})\int_0^T p^{s_{\mathrm{c}}}(\tau|R_{\mathrm{c}})d\tau \;+\; p^{s_{\mathrm{c}}}(T|R_{\mathrm{c}})\int_0^T p^{c_{\mathrm{m}}}(\tau|R_{\mathrm{m}})d\tau \tag{5}$$

This is equivalent to the derivative of the product of the color and motion cumulative distributions:

$$p_{\text{parallel}}^{s_{\mathrm{m}},s_{\mathrm{c}}}(T|R_{\mathrm{m}},R_{\mathrm{c}}) = \frac{d}{dT}[P^{s_{\mathrm{m}}}(T|R_{\mathrm{m}})P^{s_{\mathrm{c}}}(T|R_{\mathrm{c}})] \tag{6}$$

where uppercase $P$ indicates cumulative probability.

Besides the decision-time, there are sensory, motor and processing delays that contribute to the total RT. We assume that the combined non-decision latencies, $T_{\mathrm{nd}}$ are normally distributed with a mean of $\mu_{\mathrm{nd}}$ and a standard deviation of $\sigma_{\mathrm{nd}}$. The observed RT distribution for each stimulus condition and choice is obtained by convolving the distributions of the double-decision times $p^{s_{\mathrm{m}},s_{\mathrm{c}}}$ and the distribution of non-decision times, which follows from the assumption that decision and non-decision times are additive and independent. This holds for both serial and parallel models.

To avoid over-fitting, our strategy for comparing the serial and parallel models was to fit all parameters using the subset of trials in which one of the two stimulus dimensions had maximum strength (*Figure 1B*). We used the Bayesian Adaptive Direct Search method (*Acerbi and Ma, 2017*) to search over the space of parameters. The best-fitting parameters are shown in *Appendix 1—table 1*, for each participant and model type. From the marginal distributions, we predicted the double-decision choice probabilities and RTs for all combinations of motion and color coherence. To compare serial and parallel models, we used the model likelihoods of observing the choice-RT data that was not used for fitting. Because the two models have the same number of parameters (12, comprising 5 for $\Phi_{\mathrm{m}}$, 5 for $\Phi_{\mathrm{c}}$, and 2 for $\mu_{\mathrm{nd}}$ & $\sigma_{\mathrm{nd}}$), we directly compare the raw likelihoods (*Figure 2—figure supplement 1*).

We conducted a model recovery exercise to verify that our fitting procedure would recover the correct model if the data were generated by either the serial or the parallel model. For each participant and model type (serial/parallel), we generated a synthetic data set with the same number of trials per condition (combination of color and motion coherence) as completed by the participant. The parameters used to generate the synthetic data set were those that best fit the participants' data (that is, those shown in *Appendix 1—table 1*). We conducted the model comparison, just as we did for the participants' data, and assessed whether it favored the model that was used to generate the simulated data. *Figure 2—figure supplement 1* shows that our model comparison procedure can reliably identify the correct model for all 38 comparisons.

Except for Experiment 5, model fits were to all trials—correct, error and 0% coherence. For simplicity, plots of mean RT show only non-error trials (i.e. correct on both stimulus dimensions, treating either choice as correct for 0% coherence). The associated curve fits for serial and parallel models are derived from the drift-diffusion models using the trials in which the choice matches the sign of the coherence. Error trials are typically longer, as explained by the symmetrically collapsing bounds. We do not plot the error trials simply to reduce the complexity of the graphs, which would require approximately twice the number of symbols and curves. The fits and model comparisons are based on all trials. In theory, for 0% coherence both choices should have identical distributions, because the process is pure diffusion to symmetric bounds. In practice, participants typically exhibit a bias for one direction and color, and these are best approximated by a change in drift rate (e.g., $s_{\mathrm{m}}$, *Equation 2*). This can produce a systematic underestimate of the mean RT when one of the two stimulus strengths is 0% coherence, owing to the inclusion of what are effectively slow errors in the data. The degree of the underestimate depends on the shape of the collapsing bounds and degree of bias. Two exceptions to this plotting convention are noted. In *Figure 3A* (Experiment 2), all trials contribute to the mean RTs; these data are not fit by a drift-diffusion model. In *Figure 6* (Experiment 4), all trials contribute to the mean RT, and the fits are generated from simulations that include all trials, including errors. We reach identical conclusions if we restrict the analysis to correct choices.

For the binary-response (*same-different*) choice-RT task (Experiment 5), we fit both serial and parallel models using a simple proportional-rate, drift-diffusion model (e.g., *Palmer et al., 2005*). For each subject, we fit the choice-RT data from both the 1D and 2D tasks simultaneously. For the 1D task, this is a parsimonious model that uses only four degrees of freedom, $\{\kappa_{\mathrm{m}}, s_{\mathrm{m}}^0, T_{\mathrm{nd}}^{\mathrm{1D}}, B_0^{\mathrm{1D}}\}$, where

the first two terms are as defined above, $T_{\mathrm{nd}}^{\mathrm{1D}}$ is the mean non-decision time, and $\pm B_0^{\mathrm{1D}}$ is the height of the upper and lower terminating bounds. The model explains the mean RT of correct up and down choices in the 1D task. For the 2D trials, we assumed that participants applied the same decision process to each stimulus to determine an up-down choice and that the same-difference response was made by comparing the two decisions. We assumed that sensitivity ($\kappa_{\mathrm{m}}$) and any up-down bias ($s_{\mathrm{m}}^0$) was the same for the 1D and 2D choices but allowed a different bound ($B_0^{\mathrm{2D}}$) and non-decision time ($T_{\mathrm{nd}}^{\mathrm{2D}}$).

The application of stationary (i.e. non-collapsing) bounds fails to account for the distribution of RTs and it underestimates the mean RT on errors (*Ratcliff and Rouder, 1998*; *Drugowitsch et al., 2012*). The fits to the RT data from the 1D blocks (up vs. down) included only correct trials. For the same-different decisions, the up-down decisions on the two patches are not indicated. We therefore fit the mean RTs for the correct same-different choices, assuming negligible contribution of double errors (i.e. incorrect direction decisions for both the left and right patch) to the mean RT. The fit maximized the likelihood of the choice assuming binomial error (from the model) and Gaussian error for the mean RTs (from the data).

## Empirical analysis of double-decision reaction times under serial and parallel rules

We pursued a second approach to compare serial and parallel integration strategies, focusing specifically on the decision times. Unlike the fits to choice-RT, this method uses each participant's choices as ground truth. It considers only the distribution of double-decision RTs and attempts to account for them under serial and parallel logic. Instead of diffusion models, we attempt to explain the empirical distribution of double-decision RTs from unobserved probability density functions associated with the component color and motion decision times. Specifically, for each motion strength and choice ($s_{\mathrm{m}}$ and $R_{\mathrm{m}}$) and each color strength and choice ($s_{\mathrm{c}}$ and $R_{\mathrm{c}}$) we modeled the 1D decision time distributions as a gamma distribution (two parameters governing mean and standard deviation). These 1D distributions allowed us to predict the decision time on 2D trials under a serial (additive) and parallel (max) rule. The non-decision times were also modeled as four gamma distributions, one for each combination of the four choices ($R_{\mathrm{m}}$ and $R_{\mathrm{c}}$). The RT distribution was obtained by convolution of the double-decision time and non-decision time distribution. Each participant's data was fit under the serial and parallel model by maximum likelihood (using Matlab fmincon). For robustness, only combinations of strengths and choices with more than 10 trials were included in the fit. The analysis is therefore heavily weighted toward correct trials. Comparison of models was based on log likelihoods of the data given the fitted parameters for each participant. We validated this method on synthetic data from a parallel and serial simulation and showed that model recovery was accurate (*Figure 2—figure supplement 3*).

We also deployed the fit-predict strategy used in *Figure 2*, where we estimated the gamma distributions for the 1D decision times and using only the conditions in which one or the other stimulus dimension was at its maximum strength (*Figure 2—figure supplement 5*).

For the binary response task (Experiment 5), a simplified version of this model was used (*Figure 7—figure supplement 1*). Only unsigned coherence levels of each motion stimulus were considered to fit the marginal gamma distributions (i.e. combined across direction). Additionally, only RTs from correct trials were included in this model. Finally, in order to estimate the distribution of $T_{\mathrm{nd}}$, only a single gamma distribution was fitted.

## Variable duration model (Experiment 3)

We assume that when the duration of the color-motion stimulus is controlled by the experimenter, the choices are still governed by bounded integration. Thus, decisions can terminate (e.g. at time $T_{\mathrm{m}}$ for motion) before the stimulus duration, $T_{\mathrm{dur}}$ (*Kiani et al., 2008*). For example, in a 1D decision about motion stimulus with strength $s_{\mathrm{m}}$, the choice is determined by (1) the distribution of termination times, $f_+^{s_{\mathrm{m}}}(T_{\mathrm{m}})$ and $f_-^{s_{\mathrm{m}}}(T_{\mathrm{m}})$, at the positive and negative bounds, respectively, up to $T_{\mathrm{dur}}$ and (2) the probability that the sign of the unabsorbed $V_{\mathrm{m}}(t = T_{\mathrm{dur}})$ is of the corresponding sign. For example, the probability of rightward decision for a stimulus duration $T_{\mathrm{dur}}$ is

$$p(R_{\mathrm{m}} = 1) = p\{V_{\mathrm{m}}(T_{\mathrm{dur}}) > 0\} + \int_0^{T_{\mathrm{dur}}} f_+^{s_{\mathrm{m}}}(T_{\mathrm{m}} = t)dt \tag{7}$$

Note that $f_+^{s_{\mathrm{m}}}$ is not a proper density; the total probability at $t = T_{\mathrm{dur}}$ comprises absorption times at both bounds and the probability of unterminated $V_{\mathrm{m}}(T_{\mathrm{dur}})$.

To fit the data in *Figure 4* (Experiment 3) we employ two drift diffusion models, for color and motion, which only interact in the way they access the stream of sensory evidence. This interaction is governed by two parameters, one that determines the amount of time ($T_{\mathrm{buf}}$) for which processing occurs in parallel before proceeding to a serial processing stage, and a second ($p_{\mathrm{m}}^{\mathrm{1st}}$) that determines the probability that motion is prioritized over color during the serial stage. If motion is prioritized on a particular trial, for example, the motion process accumulates evidence in the serial phase until a decision bound is crossed, at which point color evidence continues to accumulate. Therefore, if $V_{\mathrm{m}}$ does not reach a decision bound before the sensory stream terminates, no further color evidence is accumulated after the parallel phase.

To model the double-decisions, we used the two 1D processes to specify the duration of the stimulus that was used for motion processing, $T_{\mathrm{m}}$, and color processing, $T_c$. On a trial in which motion is prioritized, the time component that contributed to the motion accumulation ($T_{\mathrm{m}}$) is either the time, $T_{\mathrm{m}}$, that $V_{\mathrm{m}}$ reaches a termination bound or $T_{\mathrm{dur}}$ if it does not reach a bound (in which case there is no $T_{\mathrm{m}}$). These two possibilities bear on the maximum time available for color processing ($t_{\max}^c$):

$$t_{\max}^c = \begin{cases} T_{\mathrm{dur}} & \text{if } T_{\mathrm{dur}} \leq T_{\mathrm{buf}} \text{ or } T_{\mathrm{m}} \leq T_{\mathrm{buf}} \\ T_{\mathrm{buf}} + (T_{\mathrm{dur}} - T_{\mathrm{m}}) & T_{\mathrm{m}} > T_{\mathrm{buf}} \\ T_{\mathrm{buf}} & \text{if } T_{\mathrm{dur}} > T_{\mathrm{buf}} \text{ and } \nexists T_{\mathrm{m}} \end{cases} \tag{8}$$

The three conditions in *Equation 8* can be understood intuitively. (1) If the stimulus is shorter than the parallel phase or if motion has terminated in this phase, then the maximum time available for color processing is the full duration of the stimulus. (2) If motion terminates in the serial phase then the maximum time available for color is the duration of the parallel phase and what time remains of the serial phase after motion has terminated. (3) If motion does not terminate, then color is only processed during the parallel phase. With probability $1 - p_{\mathrm{m}}^{\mathrm{1st}}$, color is prioritized, and the complementary logic holds.

Note that if $T_{\mathrm{buf}} = 0$, the model is purely serial with one change from motion to color with probability $p_{\mathrm{m}}^{\mathrm{1st}}$ or from color to motion with probability $1 - p_{\mathrm{m}}^{\mathrm{1st}}$. Although realized as a single switch, the model is qualitatively indistinguishable from other alternation schedules that preserve the same competition for processing time. For $T_{\mathrm{dur}} \leq T_{\mathrm{buf}}$, the model is effectively parallel. We fit a parallel model to the data (*Figure 4—figure supplement 2*) by fixing $T_{\mathrm{buf}}$ to the longest duration tested (1.2 s).

Each of the 1D diffusions were modeled similar to those used for the RT task, except for the following minor modifications. (1) We did not include a parameter for non-decision times, because we only modeled choices. (2) We parameterized the bound as an exponential function that is clipped to have a maximum at $u_{\mathrm{m}}$ and to start decreasing from $t = g_{\mathrm{m}}$ with a half-life of $d_{\mathrm{m}}$:

$$B_{\mathrm{m}}(t) = u_{\mathrm{m}} \min(2^{-(t - g_{\mathrm{m}})/d_{\mathrm{m}}}, 1) \tag{9}$$

with lower bound simply $-B_{\mathrm{m}}(t)$. The same parameterization applies to the color bound (terms with subscript c in *Appendix 1—table 5*).

The model was implemented in PyTorch (*Paszke et al., 2019*) with an Adam optimizer (*Kingma and Ba, 2014*) and a modified version of the cyclical learning rate schedule that simply switched back and forth between 0.05 and 0.025 every 25 epochs (*Smith, 2015*). We verified that this procedure reliably recovers the $T_{\mathrm{buf}}$ (see *Figure 4—figure supplement 1*). Briefly, in Adam, the learning rate gives an approximate upper bound to the change each parameter takes per epoch, and the step size is also adapted for individual parameters based on the running estimates of the first and second moments of the gradient. That is, a high learning rate updates parameters fast and a low learning rate allows better convergence at the expense of speed. We fit the model separately for each $T_{\mathrm{buf}}$ in steps of 40 ms from 0 to 240 ms and then in steps of 120 ms up to 1200 ms, the longest duration of the stimulus we used. The coarser sampling is justified because the model predicts decision terminations with decreasing density at longer durations, and hence the log likelihood of

the choices depends less on the buffer duration when it is long. The reported estimate of $T_{\mathrm{buf}}$ is the sample value with maximum log likelihood.

To evaluate the validity of the estimates of buffer capacity ($T_{\mathrm{buf}}$) shown in *Figure 4*, we performed two types of analyses for each participant (*Figure 4—figure supplement 1*). The first approximates the specificity, the second the sensitivity of the estimates. (1) We used the parameters of the best fitting diffusion models to the data in *Figure 4* (solid curves; see *Appendix 1—table 5*) to simulate synthetic data using a buffer duration of $T_{\mathrm{buf}} = 80$ ms. We fit the synthetic data with models with the buffer capacity fixed to other values (from 0 to 240 ms in steps of 40 ms, and from 240 to 1200 ms in steps of 120 ms). We then compared log likelihood of those fits with that of the 80 ms buffer model, and repeated the simulation 12 times. (2) We used the parameters of the best fitting diffusion models to the data in *Figure 4* to simulate synthetic data using the buffer durations, $T_{\mathrm{buf}} \neq 80$ ms, and compared two fits: with $T_{\mathrm{buf}} = 80$ ms or the simulated value.

## Multi-switch model (Experiment 4)

In the serial phase of the 2D task, the motion and color processes alternate. Experiments that provide only one RT to report both decisions allow us to estimate the overall prioritization of one stream over the other but not the frequency of alternation. In contrast, the bimanual task provides two RTs on each trial. This allows us to estimate the frequency of alternation between stimulus dimensions by fitting a model with multiple switches to the response times of the first decision in the bimanual task (Experiment 4).

The fitting was carried out in two steps. First, we fit the serial model described in *Equation 4* to the second response in the bimanual task. The parameters that best fit the data are shown in *Appendix 1—table 1*. Second, with the serial model parameters fixed, we used three additional parameters to account for the RTs to the decision that was reported first. The three parameters are: $\tau_\Delta$, controlling the average time between alternations of color and motion; $p_{\mathrm{m}}^{\mathrm{1st}}$, the probability of starting with motion; and $T_{\mathrm{nd}}^{\mathrm{1st}}$, the expectation of the non-decision time for the first response.

The alternations are modeled as a renewal. The intervals are independent and identically distributed (*iid*) as

$$f_{\mathrm{int}}(t) = \max[a, b], \tag{10}$$

where $a$ and $b$ are draws from an exponential distribution with mean $\tau_\Delta$. The expectation of the interval is

$$\mathbb{E}[f_{\mathrm{int}}(t)] = 1.5\tau_\Delta \tag{11}$$

We chose this parameterization so that the distribution of inter-switch intervals has a single peak and the max operation reduced the probability of very short intervals.

Because there is no closed-form solution to the multi-switch model, we used simulations to fit the model parameters to each participants' data. For fitting, we simulate the model 1000 times for each unique combination of color and motion strengths. From the simulations, we average the RTs for the first decisions split by whether motion or color was reported first, and binned them by both motion strength and color strength. This gives the four groupings in *Figure 6*. The parameters were fit to minimize the sum of squared-errors summed over these four groups; in other words, we minimize the sum of the squared errors for the data points shown in *Figure 6*. We used this approach rather than maximum likelihood because of the difficulties of reliably estimating the likelihood of the parameters from model simulations for continuous quantities (here, RT; *van Opheusden et al., 2020*).

## Data analysis

We used logistic regression to evaluate the influence of task type (single, double) on performance in the short-stimulus duration task (*Figure 3*). Separate regression models were fit for the color and motion decisions. The logistic regression model is:

$$logit[p_+] = \beta_0 + \beta_1 s + \beta_2 I_{\mathrm{double}} + \beta_3 s I_{\mathrm{double}} + \sum_{i}^{N_{\mathrm{subj}-1}} \beta_{3+i} I_{\mathrm{subj}} \tag{12}$$

where $p_+$ is the probability of a positive choice ('rightward' for the motion task, 'blue' for the color

task) , $s$ is (signed) stimulus strength, $I_{\text{double}}$ is an indicator variable for task type (single or double), $\beta_3$ is an interaction term which indicates how the influence of strength on choice changes in the double task relative to the single task, and $I_{\text{subj}}$ is an indicator variable that takes a value of 1 if the trial was completed by subject subj and 0 otherwise. The final term with the summation allows for the possibility that different participants had different overall choice biases.

We also used logistic regression to assess whether the strength of one stimulus dimension affected the accuracy of the other decision. Separate regression models were fit for the color and motion decisions. The logistic regression model to assess whether color strength affects motion choice is

$$logit[p_+] = \beta_0 + \beta_1 s_m + \beta_2 |s_c| + \beta_3 s_m |s_c| \tag{13}$$

where the $\beta_3$ term accommodates the possibility that the color coherence could affect the slope of the logistic function of motion coherence. We used an analogous equation to ask whether motion strength affected color sensitivity. For both logistic regression models (*Equation 12*, *Equation 13*), to test whether the interaction ($\beta_3$) has explanatory power in the model we compared the Bayesian Information Criterion (BIC) for nested regression models with and without the $\beta_3$ term. For *Equation 13* data were fit for each participant and the BICs were summed.

For the model-free analysis of the time course of the influence of motion and color information on choice *Figure 3*, we obtained choice-conditioned averages of the color and motion energies extracted from the random-dot stimuli. Because the stimulus is stochastic, the motion and color energies vary from one trial to another and also within a trial. We quantify the motion fluctuations by convolving the sequence of random dots presented in each trial $i$, $S_i(x, y, t)$, with spatiotemporal nonlinear filters, $f^{\text{R}}(x, y, t)$ and $f^{\text{L}}(x, y, t)$, selective to rightward and leftward motion, respectively. These filters are specified in other publications (e.g. *Adelson and Bergen, 1985*; *Kiani et al., 2008*). The results of the convolutions, $H^{\text{R}}(x, y, t)$ and $H^{\text{L}}(x, y, t)$ are combined over space to obtain the motion energy for each direction, as a function of time $R_i(t)$ and $L_i(t)$. The net motion energy is the difference of the right and left signals, detrended by the average over all trials with the same motion strength and direction (i.e., the signed coherence, $s_{\text{m}}$):

$$M_i(t) = R_i(t) - L_i(t) - \langle R(t) - L(t) \rangle_{i \in s_{\text{m}}} \tag{14}$$

where $\langle \cdots \rangle$ connotes the mean over all trials sharing the same signed coherence. $M_i(t)$ represents the residual fluctuation of motion energy caused by the stochastic nature of the stimulus on that trial. The four traces in *Figure 3B* (*top*) are averages of the residuals from all motion strengths and directions, grouped by whether the choice was rightward or leftward in the 1D and 2D tasks.

We performed a similar analysis to extract the color energy from the stimulus. We calculated the difference between the number of blue and yellow dots shown on each video frame. We filtered this discrete representation by convolution with the impulse response function shown in *Figure 3B* (*bottom*). We constructed residual *excess color for blue*, $C_i(t)$ analogous to $M_i(t)$ in *Equation 14*. The four traces in *Figure 3B* (*middle*) are averages of the residuals from all trials, grouped by whether the choice was blue or yellow in the 1D and 2D tasks. Note that the blurring step does not affect the conclusions drawn from this analysis. The unfiltered color residuals also support the conclusion that the weighting of evidence samples used to form color decisions (in Experiment 2) was the same for 1D and 2D trials.

### Data and code availability

Data are available on the figshare repository (*Kang et al., 2021a*). Code is available on the GitHub repository https://github.com/yulkang/2D_Decision; (*Kang et al., 2021b*; *Kang et al., 2021c* copy archived at swh:1:rev:91922907c5ecaa832bdc6ee6cb285095905f4cac).

## Acknowledgements

We thank Daphna Shohamy and Mariano Sigman for contributions to the theoretical underpinnings of our study, and we thank Stanislas Dehaene, Gabriel Stine, Naomi Odean, S Shushruth, and Aniruddha Das for comments on an earlier draft of the manuscript.

## Additional information

### Funding

| Funder | Grant reference number | Author |
|---|---|---|
| National Eye Institute | T32EY01393 | Yul HR Kang |
| Simons Foundation | 414196 | Danique Jeurissen |
| Brain and Behavior Research Foundation | 28476 | Danique Jeurissen |
| Howard Hughes Medical Institute | | Michael N Shadlen |
| National Eye Institute | R01EY11378 | Michael N Shadlen |
| National Institute of Neurological Disorders and Stroke | R01NS113113 | Michael N Shadlen |

The funders had no role in study design, data collection and interpretation, or the decision to submit the work for publication.

### Author contributions

Yul HR Kang, Conceptualization, Data curation, Software, Formal analysis, Validation, Investigation, Visualization, Methodology, Writing - original draft, Writing - review and editing; Anne Löffler, Conceptualization, Data curation, Software, Formal analysis, Validation, Investigation, Visualization, Methodology, Writing - original draft, Project administration, Writing - review and editing; Danique Jeurissen, Data curation, Software, Methodology, Project administration, Conceptualization, Investigation, Writing - original draft, Writing - review and editing; Ariel Zylberberg, Software, Formal analysis, Supervision, Validation, Visualization, Methodology, Writing - original draft, Writing - review and editing; Daniel M Wolpert, Conceptualization, Resources, Software, Supervision, Validation, Methodology, Writing - original draft, Project administration, Writing - review and editing; Michael N Shadlen, Conceptualization, Resources, Software, Supervision, Funding acquisition, Validation, Visualization, Methodology, Writing - original draft, Project administration, Writing - review and editing

### Author ORCIDs

Yul HR Kang ![ORCID] https://orcid.org/0000-0002-8846-5296
Anne Löffler ![ORCID] https://orcid.org/0000-0001-9086-1290
Danique Jeurissen ![ORCID] https://orcid.org/0000-0003-3835-5977
Ariel Zylberberg ![ORCID] https://orcid.org/0000-0002-2572-4748
Daniel M Wolpert ![ORCID] https://orcid.org/0000-0003-2011-2790
Michael N Shadlen ![ORCID] https://orcid.org/0000-0002-2002-2210

### Ethics

Human subjects: The study was approved by the local ethics committee (Institutional Review Board of Columbia University Medical Center IRB-AAAL0658 & IRB-AAAR9148). Thirteen participants (5 male and 8 female, age 23-40, median = 26, IQR = 25-32, mean = 28.3, SD = 5.74) provided written informed consent and took part in the study.

### Decision letter and Author response

Decision letter https://doi.org/10.7554/eLife.63721.sa1
Author response https://doi.org/10.7554/eLife.63721.sa2

## Additional files

### Supplementary files

• Transparent reporting form

## Data availability

The data is on figshare at: https://dx.doi.org/10.6084/m9.figshare.13607255. The code is available at the following repository: https://github.com/yulkang/2D_Decision (copy archived at https://archive.softwareheritage.org/swh:1:rev:91922907c5ecaa832bdc6ee6cb285095905f4cac/). The figshare (allows deposition of big data) and github (suitable for maintenance of code) repositories refer to each other.

The following dataset was generated:

| Author(s) | Year | Dataset title | Dataset URL | Database and Identifier |
|---|---|---|---|---|
| Kang YHR, Löffler A, Jeurissen D, Zylberberg A, Wolpert DM, Shadlen MN | 2021 | Data for "Multiple decisions about one object involve parallel sensory acquisition but time-multiplexed evidence incorporation" | https://dx.doi.org/10.6084/m9.figshare.13607255 | figshare, 10.6084/m9.figshare.13607255 |

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

# Appendix 1

**Appendix 1—table 1.** Parameter values for the best-fitting serial model (Experiments 1 and 4). Note that the rate of collapse parameters ($a_m$ and $a_c$) are limited to a maximum of 10 (an almost instantaneous bound collapse) and the time of the start of the collapse ($d_m$ and $d_c$) are limited to 4 s.

| Task | Subj. | $\kappa_m$ | $u_m$ | $a_m$ | $d_m$ (s) | $s_m^0$ | $\kappa_c$ | $u_c$ | $a_c$ | $d_c$ (s) | $s_c^0$ | $\mu_{nd}$ (s) | $\sigma_{nd}$ (s) |
|---|---|---|---|---|---|---|---|---|---|---|---|---|---|
| Eye RT | 1 | 9.97 | 0.98 | 6.45 | 3.47 | −0.02 | 5.77 | 0.83 | 10 | 4 | −0.01 | 0.3 | 0.001 |
| | 2 | 21.99 | 0.99 | 3.65 | 2.52 | 0.01 | 11.39 | 0.68 | 10 | 2.61 | 0.02 | 0.35 | 0.002 |
| | 3 | 39.25 | 0.83 | 9.91 | 3.26 | −0.01 | 6.29 | 3.62 | 1.23 | −0.19 | 0 | 0.31 | 0.004 |
| Unimanual | 6 | 14.02 | 1.3 | 1.97 | 3.96 | 0.01 | 7.24 | 0.91 | 3.09 | 2.18 | 0.04 | 0.34 | 0.001 |
| | 7 | 13.94 | 1.42 | 2.01 | 3.6 | −0.01 | 4.97 | 1.33 | 1.21 | 2.79 | 0.06 | 0.41 | 0.001 |
| | 8 | 9.98 | 1.11 | 4.07 | 4 | 0 | 7.16 | 1.03 | 2.15 | 2.93 | 0.02 | 0.31 | 0.001 |
| | 9 | 12.84 | 0.75 | 10 | 2.64 | 0 | 7.18 | 0.91 | 10 | 3.28 | 0.02 | 0.69 | 0.08 |
| | 10 | 20.81 | 0.88 | 10 | 4 | −0.01 | 7.46 | 0.91 | 3.72 | 2.14 | −0.06 | 0.37 | 0.001 |
| | 11 | 19.56 | 0.84 | 4.81 | 2.72 | 0 | 7.67 | 1.73 | 0.32 | 0.51 | −0.03 | 0.42 | 0.004 |
| | 12 | 12.05 | 0.96 | 2.51 | 3.98 | 0 | 5.28 | 0.96 | 10 | 3.35 | 0.03 | 0.44 | 0.001 |
| | 13 | 13.13 | 1 | 2.67 | 4 | −0.02 | 5.87 | 1.06 | 0.98 | 3.06 | 0.02 | 0.41 | 0.001 |
| Bimanual | 6 | 8.39 | 0.74 | 10 | 3.44 | 0 | 4.36 | 0.98 | 10 | 4 | 0.04 | 0.33 | 0.001 |
| | 7 | 9.45 | 0.8 | 10 | 4 | −0.02 | 4.56 | 0.97 | 7.67 | 3.92 | 0.05 | 0.3 | 0.001 |
| | 8 | 11.89 | 1.44 | 0.65 | 1.64 | −0.04 | 6.9 | 1.03 | 1.78 | 1.85 | −0.02 | 0.37 | 0.007 |
| | 9 | 13.57 | 0.87 | 10 | 4 | 0 | 7.05 | 0.86 | 4.57 | 2.49 | −0.1 | 0.44 | 0.002 |
| | 10 | 13.03 | 1.34 | 1.51 | 3.84 | 0 | 6.9 | 1.07 | 2.1 | 2.81 | 0.07 | 0.38 | 0.022 |
| | 11 | 12.56 | 1.68 | 0.77 | 2.35 | 0 | 6.78 | 0.95 | 2.32 | 3.05 | 0.05 | 0.46 | 0.001 |
| | 12 | 12.65 | 1.03 | 6.16 | 3.33 | 0.01 | 6.01 | 1.08 | 10 | 3.96 | −0.03 | 0.31 | 0.001 |
| | 13 | 8.91 | 1.16 | 5.14 | 3.91 | 0.01 | 4.25 | 1.05 | 1.42 | 3.87 | −0.09 | 0.3 | 0.001 |

**Appendix 1—table 2.** Parameter values for the best-fitting switching model (Experiment 4–bimanual).

| Subj. | $\tau_\Delta$ (s) | $p_m^{1st}$ | $T_{nd}^{1st}$ (s) |
|---|---|---|---|
| 6 | 1.48 | 0.5 | 0.42 |
| 7 | 0.88 | 0.33 | 0.59 |
| 8 | 8.73 | 0.86 | 0.38 |
| 9 | 1.27 | 0.88 | 0.43 |
| 10 | 0.16 | 0.05 | 0.38 |
| 11 | 0.18 | 0.86 | 0.71 |
| 12 | 0.22 | 0.6 | 0.5 |
| 13 | 0.74 | 0.92 | 0.64 |

**Appendix 1—table 3.** Parameter values for the best-fitting drift-diffusion model (Experiment 5–binary-choice).

| Task | Subj. | $\kappa_m$ | $B_0^{1D}$ | $B_0^{2D}$ | $s_m^0$ | $T_{nd}^{1D}$ (s) | $T_{nd}^{2D}$ (s) |
|---|---|---|---|---|---|---|---|
| Binary choice | 7 | 8.96 | 0.85 | 1.09 | 0.03 | 0.35 | 0.24 |
| | 12 | 6.83 | 1.17 | 1.03 | −0.04 | 0.43 | 0.35 |

**Appendix 1—table 4.** Sensitivity of color and motion choices obtained on single decision and double-decision tasks.

*Eye-RT*. Sensitivity is the slope of a logistic fit to the proportion of rightward (blue) choices as a function of motion (color) strength. Values in the 1D columns are obtained from different blocks in which participants were instructed to answer only the motion direction or color dominance. Values in the 2D columns are obtained from the double-decisions, using either the motion or color choice. *Binary choice*. Here, the 2D task refers to the same-different task. Sensitivity for the 1D task is the slope of a logistic fit to direction choices for one patch (ignoring the other). For the same-different task, the direction choices on individual patches is not reported. The 1D sensitivity is estimated by fitting the proportion of same and different choices assuming the same sensitivity to motion direction for each patch. Parentheses show s.e.

| Task | Subj. | Motion sensitivity | | Color sensitivity | |
|------|-------|---------|---------|---------|---------|
| | | **1D task** | **2D task** | **1D task** | **2D task** |
| Eye-RT | S1 | 17.5 (3.0) | 18.7 (0.8) | 10.6 (0.7) | 9.5 (0.5) |
| | S2 | 43.4 (10.9) | 48.1 (2.1) | 15.8 (0.7) | 16 (0.5) |
| | S3 | 59 (2.0) | 68.2 (1.8) | 10.5 (0.4) | 11.5 (0.3) |
| Binary choice | S7 | 13.3 (1.0) | 13.9 (1.0) | | |
| | S12 | 13.2 (0.9) | 19.3 (1.6) | | |

**Appendix 1—table 5.** Parameter values for the best-fitting buffer + serial model (Experiment 3–Variable-duration stimulus presentation).

| Task | Subj. | $\kappa_m$ | $u_m$ | $g_m$ (s) | $d_m$ (s) | $s_m^0$ | $\kappa_c$ | $u_c$ | $g_c$ (s) | $d_c$ (s) | $s_c^0$ | $p_m^{1st}$ | $T_{buf}$ (s) |
|------|-------|-----------|-------|----------|----------|--------|-----------|-------|----------|----------|--------|------------|-------------|
| VD | 4 | 22.29 | 0.86 | 0.47 | 1.19 | 0.01 | 9.18 | 1.39 | 0.18 | 0.27 | −0.01 | 0.80 | 0.08 |
| | 5 | 9.60 | 0.99 | 0.68 | 0.40 | 0.01 | 12.11 | 0.74 | 0.11 | 0.11 | 0.04 | 0.96 | 0.08 |

