## [Decision Letter]

**Acceptance summary:**

Kang, Shadlen and colleagues investigate the processes of evidence accumulation and evaluation when more than one decision about an object is required in order to arrive at a correct response. The two perceptual decisions are about the direction of motion (left/right) and the dominant color of the dots (blue/yellow) of a random-dot kinematogram. This systematic psychophysical study is underpinned by a full drift diffusion model that suggests that while the two aspects of the stimulus are acquired in parallel, the evidence is integrated serially into the decision process. The authors makes a persuasive case that when a decision has to be made about two feature dimensions of a stimulus, processing is subject to a bottleneck that allows only a limited amount of information from one dimension to be passed on to a decision stage. The methods are detailed and the experiments are rigorous and convincing. The complex set of experimental results is impressively fitted with a single model. This is a careful study of general significance for our understanding of the timescales of decision processes involved in cognition.

**Decision letter after peer review:**

Thank you for submitting your article "Multiple decisions about one object involve parallel sensory acquisition but time-multiplexed evidence incorporation" for consideration by *eLife*. Your article has been reviewed by three peer reviewers, one of whom is a member of our Board of Reviewing Editors, and the evaluation has been overseen by Tirin Moore as the Senior Editor. The following individuals involved in review of your submission have agreed to reveal their identity: Pascal Mamassian (Reviewer #2); Preeti Verghese (Reviewer #3).

The reviewers have discussed the reviews with one another and the Reviewing Editor has drafted this decision to help you prepare a revised submission.

Summary:

The manuscript "Multiple decisions about one object involve parallel sensory acquisition but time-multiplexed evidence incorporation" by Shadlen and colleagues investigates the processes by which evidence is accumulated and evaluated when more than one decision about an object has to be made in order to make a correct response. The visual stimulus is a random-dot kinematogram and the two decisions are about the direction of motion (left/right) and the dominant color of the dots (blue/yellow). This systematic psychophysical study is underpinned by a full drift diffusion model that suggests that while the two aspects of the stimulus are acquired in parallel, the evidence is integrated serially into the decision process. The manuscript makes a persuasive case that when a decision has to be made about two feature dimensions of a stimulus, processing is subject to a bottleneck that allows only a limited amount of information from one dimension to be passed on to a decision stage. The methods are detailed and the experiments are rigorous and convincing. The complex set of experimental results is impressively fitted with a single model.

This is a careful study of general significance for our understanding of the timescales of decision processes involved in cognition.

The manuscript is quite long and would benefit from a more stringent organisation, some additional information about task and set-up, as well as more consideration of the processes downstream from the computation of the decision variable.

Essential revisions:

1) All three referees found the manuscript lengthy, particularly in results and discussion, and therefore difficult to read. It would be very helpful to streamline the different result sections for each individual experiment to allow easier cross-reference and for both results and discussion to focus on the key findings. As a consequence of the complex text structure and meandering explanations, we found it at times difficult to piece all the relevant task and analysis information together.

Here are a few key suggestion for re-organization and shortening:

a) There are a lot of different experiments described in the text for which it is difficult to find the related methods in the Materials and methods section (e.g. which experiment in the main text corresponds to the "Choice-reaction time task (eye)"?).

Maybe as well as a consistent name for each experiment in Results, Figures, Tables and Materials and methods, the different experiments could also be numbered in their treatment in the paper – the names are not necessarily intuitive for the reader who needs to cross reference to the Materials and methods.

b) For the Results in particular, it would facilitate reading if there was a concise section near the beginning of each psychophysical experiment that stated briefly number of participants, number of trials (and sessions) per subject, task, visual stimulus parameters, response effector, what randomisation and controls. For some of the first experiments described, one has to go through results, figure legends and methods to piece all this together.

c) It is important to systematically compare the results from experiments where observers performed the double task (what the authors call "2D") to experiments with a single task ("1D"). This "1D" condition gives the baseline for both sensitivity and reaction times measures. It seems that the authors have run the "1D" condition sometimes (see e.g. Figure 3), but we were not convinced that they have run it in all the experiments. When available, we think the results from this "1D" condition should be systematically presented (e.g. in dashed lines in Figure 2). Ultimately, is it possible to test their model by only fitting the "1D" data and predict the "2D" results?

d) The choice and response time experiments for eyes and hands, the brief duration stimulus, the variable duration stimulus, and the bimanual task systematically build up the case for a parallel information acquisition stage and a serial decision stage. However, we think that the binary response with the double-decision does not add so much to the evidence. This task may have an additional stage of comparing the two motion stimuli to decide whether they are the same or different, which might introduce additional elements not present in the independent decisions associated with judging the motion and color aspects of a single stimulus.

Maybe this or some other experiment can be moved to Supplementary Materials so that the main text focuses on the essential aspects of this nice work.

e) The Discussion is quite complex without a clear organisation. Is it really necessary to have a 6-page discussion? Could the key points be made more concisely and a clearer organisation, for instance with a small number of subheadings or clear sign posts be imposed?

2) It is not clear to us why the response time data are fit with γ distributions, when the drift diffusion model can handle both choice and response time. Removing the additional and unnecessary γ distribution fits will help focus the paper.

3) One piece of information that is important for this type of psychophysical experiment are the actual instructions that subjects received before/when they were doing each of the experiments. I assume this happened with a training task, on the computer screen or through an information sheet to ensure that participants carried out the same cognitive task? We are sorry if we missed this.

The specific concern is to what extent participants were guided to make a combined or serial judgements. For instance, were the participants instructed to "respond upper right for yellow dots moving right" OR "move your eyes to one of the right targets for rightwards moving dots, choose the upper target for yellow".

Could the authors include the precise wording for each experiment in the Materials and methods, please? If the instruction was of a serial nature for colour and motion, this issue needs to be discussed in more depth.

4) We were concerned whether motor preparation and issues for the motor response were ignored by the authors.

a) The authors argue that there is an initial 0.1s when both stimulus dimensions are acquired in parallel. I did not find a motor execution part in the non-decision time in the drift-diffusion model of the authors. What is the evidence that the 0.1s is purely sensory and not partly motor?

b) Areas in parietal and prefrontal cortex that have been proposed to compute/represent the decision variable show persistent activity, necessary for temporarily buffering information – rather than most visual cortical areas, in particular with regards to visual motion areas. How do the authors exclude that the bottleneck is not downstream from the computation of the decision variable(s), for instance by the process that integrates two decision variables in one motor response?

More generally how do the authors account for motor preparation time in their model?

5) Serial vs Parallel model: The serial model does a much better job than the parallel model of describing the response time data in Figure 1 Figure 2, Figure 3, Figure 4 and Figure 5. However, there is no discussion of the systematic underestimation of response times at very low values of motion and color coherence.

6) Eye vs. Hand: Why are participants less sensitive to color coherence in the eye condition? Does the longer manual response time mask these differences? In addition, the discrepancies of the serial model at low coherence values are greater for the eye data (see point 3 above). How does this relate to the shorter latency of eye movements, relative to manual responses?

7) The motion tasks is a temporal task requiring the integration of information over time while the colour task is fundamentally a spatial task that could be solved using a single frame. To what extent can this explain the results the authors have obtained on differential cross-interference for the variable stimulus duration?

8) Given these are human subjects, could the serial nature of the decision process have to do with the subjects internally verbalising the individual decisions before combining? We noted in this context that even for eye movement responses reaction times are quite long, much longer than a monkey would take for similar tasks. How do the authors exclude the possibility of verbalisation playing a role in the time course?

9) In the experiment with variable stimulus duration, it seems that the different durations are interleaved in a block of trials. If so, this creates additional uncertainty for the observer. Would the authors predict the same results if only one such duration was presented in a block?

10) In the using a double decision with binary response:

a) How can we be assured that participants can execute two independent movements at roughly the same time? We thought that bimanual coordination tasks were quite difficult to master (e.g. Mechsner et al., 2001). Could the authors comment on the amount of training required to perform their task at a reasonable motor performance level?

b) We discovered that only correct trials were analysed in the reaction time data (last line of caption of Figure 7). Why is this an appropriate choice? Was this also the case for the other experiments?

c) We were wondering why the authors did not use a conjunction judgment on their original stimulus, something like "respond UP" when the stimulus is either "left-blue" or "right-yellow", and "respond DOWN" when the stimulus is either "left-yellow" or "right-blue" to "down response". Presenting two motion fields in the two hemifields seems quite different from all the other experiments.

11) Is it really reasonable to assume that evidence is retained in the buffer without any loss? If there was a loss, this would be seen in the accuracy data, so I suspect this is the reason the authors did not consider this possibility. But what is the evidence for retaining information in working memory without any loss?

12) Figure 8 is one possible instantiation of a serial bottleneck at the decision stage. But it is not clear why the transfer from the buffer has such a large delay (180ms). What is the argument against a shorter transfer time, but a longer refractory period before the decision stage can accept new information? Since this is the speculative part of the paper, perhaps it does not have to be so detailed.

13) In addition to the preference to prioritize motion, was there a tendency to prioritize the easier task, or was the motion task prioritized because observers were more sensitive to motion (at least for the eye condition)?

---

## [Author Response]

Essential revisions:1) All three referees found the manuscript lengthy, particularly in results and discussion, and therefore difficult to read. It would be very helpful to streamline the different result sections for each individual experiment to allow easier cross-reference and for both results and discussion to focus on the key findings. As a consequence of the complex text structure and meandering explanations, we found it at times difficult to piece all the relevant task and analysis information together.Here are a few key suggestion for re-organization and shortening:a) There are a lot of different experiments described in the text for which it is difficult to find the related methods in the Materials and methods section (e.g. which experiment in the main text corresponds to the "Choice-reaction time task (eye)"?).Maybe as well as a consistent name for each experiment in Results, Figures, Tables and Materials and methods, the different experiments could also be numbered in their treatment in the paper – the names are not necessarily intuitive for the reader who needs to cross reference to the Materials and methods.

We have now numbered the experiments in the revised paper in the following way:

Experiment 1. Double-decision reaction time (eye and unimanual)

Experiment 2. Brief stimulus presentation (eye)

Experiment 3. Variable-duration stimulus presentation (eye)

Experiment 4. Two-effector double-decision reaction time (bimanual)

Experiment 5. Binary-response double-decision reaction time

We have revised the methods and Results sections, as well as all figures, accordingly, to facilitate cross referencing.

b) For the Results in particular, it would facilitate reading if there was a concise section near the beginning of each psychophysical experiment that stated briefly number of participants, number of trials (and sessions) per subject, task, visual stimulus parameters, response effector, what randomisation and controls. For some of the first experiments described, one has to go through results, figure legends and methods to piece all this together.

We have now added a brief description at the beginning of each experiment in the Results section, including the number of participants/sessions/trials, coherence levels and their randomization, response effectors, and controls.

c) It is important to systematically compare the results from experiments where observers performed the double task (what the authors call "2D") to experiments with a single task ("1D"). This "1D" condition gives the baseline for both sensitivity and reaction times measures. It seems that the authors have run the "1D" condition sometimes (see e.g. Figure 3), but we were not convinced that they have run it in all the experiments. When available, we think the results from this "1D" condition should be systematically presented (e.g. in dashed lines in Figure 2). Ultimately, is it possible to test their model by only fitting the "1D" data and predict the "2D" results?

The answer to the final question is, no, it is not possible to test the model this way. The intuition for why is that the decision maker controls speed-accuracy tradeoff, and there is no reason to presume that the same setting would be applied in 1D and 2D decisions. Consider that the error rate for difficult 1D and 2D decisions is 0.5 and 0.75, respectively. As we and others have shown, reaction times are longer on 4- vs. 2-choice tasks (Churchland, Kiani and Shadlen, 2008; Usher, Olami and McClelland, 2002; Ditterich, 2010). This is why we chose to focus on the more informative comparisons: (i) difficult motion paired with easy vs. difficult color, and (ii) difficult color paired with easy vs. difficult motion. These comparisons get to the heart of the question of parallel vs. serial. Consideration of speed vs. accuracy also apply to the variable duration task (Experiment 3) because decision makers still terminate their decisions with what they deem as sufficient evidence even in this setting (Kiani, Hanks and Shadlen, 2008). However, the consideration does not apply to the experiment using very short durations (Experiment 2) because differences in termination criteria would not be expected to play a role. All difficult trials would benefit from more information in this task, and the comparisons of 1D and 2D versions of this task are informative (Figure 3).

(d) The choice and response time experiments for eyes and hands, the brief duration stimulus, the variable duration stimulus, and the bimanual task systematically build up the case for a parallel information acquisition stage and a serial decision stage. However, we think that the binary response with the double decision does not add so much to the evidence. This task may have an additional stage of comparing the two motion stimuli to decide whether they are the same or different, which might introduce additional elements not present in the independent decisions associated with judging the motion and color aspects of a single stimulus.Maybe this or some other experiment can be moved to Supplementary Materials so that the main text focuses on the essential aspects of this nice work.

We agree that the same-different task (Experiment 5) involves a comparison of the two direction decisions, but this should add a fixed amount of time because it should not depend on the difficulty of the direction decisions for each patch. Such a coherence-independent stage would be captured by the non-decision time. Like color and motion, the assessment of direction in two parts of the visual field would be expected to occur in parallel, based on what is known from neurophysiology and visual psychophysics (e.g., masking and aftereffects), but that is not what we find. Our experience presenting this material is that people find the additive decision times surprising. More importantly, this experiment addresses a potential confound by showing that it is the double decision that adds time, not the doubling of the number of possible responses. Moreover, the findings demonstrate that the serial incorporation of evidence into a double decision is not restricted to different perceptual modalities, such as color and motion. We therefore favor keeping this experiment in the main text.

(e) The Discussion is quite complex without a clear organisation. Is it really necessary to have a 6-page discussion? Could the key points be made more concisely and a clearer organisation, for instance with a small number of subheadings or clear sign posts be imposed?

We are aware of this and share the reviewers’ concern. We removed material that could be deemed tangential or non-essential or redundant, but this amounted to under one page. We added subheadings and revised the text to render the structure of the argument clearer.

(2) It is not clear to us why the response time data are fit with γ distributions, when the drift diffusion model can handle both choice and response time. Removing the additional and unnecessary γ distribution fits will help focus the paper.

The approach using γ distributions provides an empirical comparison of summation vs. max logic. It is unconstrained by the choice behavior and does not depend on the assumptions inherent in drift-diffusion models. It explains the observed double-decision RT distributions as either the serial or parallel combination of latent (i.e., unobservable) distributions of color and motion decision times. It escapes the systematic discrepancies noted in Figure 2, because its only penalty is this very discrepancy between observed and predicted RT. Yet it still favors the serial combination rule. It is also important for the comparison of parallel and serial models in Experiment 5 (binary same-different) because the parsimonious drift-diffusion model (with flat bounds) does not depict realistic distributions of decision times, and this could penalize the max and sum operations differently. Of course, the limitation to the approach is that it does not constrain the relationship between choice accuracy and decision time, so it cannot replace drift diffusion models. We have revised the manuscript in several places to make these points clearer to the reader.

(3) One piece of information that is important for this type of psychophysical experiment are the actual instructions that subjects received before/when they were doing each of the experiments. I assume this happened with a training task, on the computer screen or through an information sheet to ensure that participants carried out the same cognitive task? We are sorry if we missed this.The specific concern is to what extent participants were guided to make a combined or serial judgements. For instance, were the participants instructed to "respond upper right for yellow dots moving right" OR "move your eyes to one of the right targets for rightwards moving dots, choose the upper target for yellow".Could the authors include the precise wording for each experiment in the Materials and methods, please? If the instruction was of a serial nature for colour and motion, this issue needs to be discussed in more depth.

We have added the instructions for each experiment to the Materials and methods section. Importantly, participants were guided to make combined, rather than serial, judgments in the 2D task. For example, the instructions for “Experiment 1. Double-decision reaction time (eye)” were as follows: “Your task is to answer based on both motion and color. When the motion is right and color is yellow, the answer is top right, and when the motion is right and color is blue, the answer is bottom right.”

(4) We were concerned whether motor preparation and issues for the motor response were ignored by the authors.(a) The authors argue that there is an initial 0.1s when both stimulus dimensions are acquired in parallel. I did not find a motor execution part in the non-decision time in the drift-diffusion model of the authors. What is the evidence that the 0.1s is purely sensory and not partly motor?

The reviewer could be referring to two issues. The 0.12 s is the duration of the brief stimulus in Experiment 2 that had to be acquired in parallel. This experiment is not the usual choice-RT design because the stimulus is restricted in time. We do not apply a drift-diffusion model to these data in part because it would be unclear how to characterize the non-decision time. However, the evidence that the 0.12 s worth of sensory information has been acquired in parallel is that it is used to make the double decision. It has nothing to do with motor preparation or non-decision time, which typically combines both sensory processing delays and motor preparation, along with any other delay that does not depend on trial type (e.g., difficulty).

Alternatively, the reviewer could be referring to the estimate of the buffer in Figure 4 (Experiment 3, variable duration), which we refer to as the initial parallel phase. Perhaps the reviewer is wondering why we assume the parallel acquisition occurs at the beginning and not the end of the decision? That is supported by Experiment 2 and the psychophysical reverse correlation analyses.

We are unsure what it means to regard acquisition as “partly motor”. We are simply saying that information from both dimensions affect the decision. For this to be true, the information had to be acquired in parallel.

The revised manuscript contains a clearer explanation of the non-decision time as well as the reasoning about the parallel acquisition and buffer.

(b) Areas in parietal and prefrontal cortex that have been proposed to compute/represent the decision variable show persistent activity, necessary for temporarily buffering information – rather than most visual cortical areas, in particular with regards to visual motion areas. How do the authors exclude that the bottleneck is not downstream from the computation of the decision variable(s), for instance by the process that integrates two decision variables in one motor response?

If the bottleneck were downstream from the computation of the DV, the RTs would not be additive. The process would be coherence-independent and therefore absorbed into non-decision time. There would also be no interference effects in Experiment 3 (variable duration). The bimanual task is also relevant. We see the same seriality even when the participant does not integrate the two decisions into one motor response. From this and other concerns (e.g., #8, below), we sense that we failed to get across the point that all delays that do not vary as a function of stimulus strength are absorbed in the non-decision time. We have revised the manuscript to reinforce this point.

More generally how do the authors account for motor preparation time in their model?

It’s a component of the non-decision time (*T*_nd_) along with other coherence-independent contributions. This is now better explained in the revised manuscript. See also response to point 4b.

(5) Serial vs Parallel model: The serial model does a much better job than the parallel model of describing the response time data in Figure 1, Figure 2, Figure 3, Figure 4 and Figure 5. However, there is no discussion of the systematic underestimation of response times at very low values of motion and color coherence.

This is an important point. One possibility we considered is that switching between stimulus dimensions incurs a time penalty. We have not convinced ourselves that this is correct, and suspect that it is at best partially true. We are more confident about two other factors: (i) a constant variance assumption in the DDM fits that leads to an underestimate of RT at low motion coherences, and (ii) inclusion of all trials at 0% coherence (color and motion) that inflates the mean RTs. Regarding the first, our diffusion model is under-parameterized. In the standard DDM championed by Ratcliff, Wagenmakers and others, each stimulus strength would be assigned its own drift rate, whereas we impose a proportionality constraint. This assumes a linear relationship between the ratio of signal and variance, what amounts to an assumption of constant variance. In reality the variance at low coherences is likely to be smaller than what the model asserts (Shadlen et al., 2006; Britten et al., 1993). By freeing up this constraint, we can fit the data nicely as we have shown in other publications (Zylberberg, Fetsch and Shadlen, 2016). We elected not to do this here because (don’t laugh) we were trying to keep matters simple. Incorporating the extra degree of freedom in the parameterization of the DDM would remedy the issue, but we do not feel it is worth the added complexity since the model comparison is already decisive and it is confirmed by a an even more compelling approach (the empirical approach using γ distributions), which completely resolves the systematic error (see Figure 2—figure supplement 5). Regarding the second factor, all participants have a small bias in favor of a color and a direction. This means that just under half of the 0% decisions are effectively errors (i.e., incongruent with the drift rate). They inflate the observed mean and are not reflected in the curves. They are accounted for in the actual fits. We now discuss the systematic discrepancy in the revised manuscript.

(6) Eye vs. Hand: Why are participants less sensitive to color coherence in the eye condition? Does the longer manual response time mask these differences? In addition, the discrepancies of the serial model at low coherence values are greater for the eye data (see point 3 above). How does this relate to the shorter latency of eye movements, relative to manual responses?

We assume the reviewer is referring to the apparently shallower choice function in Figure 2A (right, top) and the corresponding function in panel B. This is explained by the different scales. The abscissae in panel-A used a scale normalized to ±1 for each participant before averaging because the three participants did not view identical motion strengths. In panel B all participants viewed the same strengths, so we plot by the true motion and color strengths. Therefore, the sensitivity parameter (k_c_) offers a more informative comparison. It is a conversion of units of color coherence into units of signal-to-noise, and it reveals little difference between 2 of the 3 “eye” participants and the 8 “unimanual” participants. We have further clarified the normalization in the figure legend and methods. (We assume by point 3 the reviewer means point 5.)

(7) The motion tasks is a temporal task requiring the integration of information over time while the colour task is fundamentally a spatial task that could be solved using a single frame. To what extent can this explain the results the authors have obtained on differential cross-interference for the variable stimulus duration?

The reviewer’s assertion is partially correct. Information about color dominance is present in all video frames, whereas the relevant information about motion direction is first presented in the fourth frame (40 ms after the first). From then on, all motion frames contribute informatively to direction. However, both tasks benefit from “integration of information over time”, as is clear from the variable duration task (Experiment 3), improving accuracy over several hundred ms. Therefore, we think the “differential cross-interference” is not explained by low level differences in color vs. motion processing but by prioritization, which is only partially addressed in our study (e.g., Experiment 4). Please see our response to point 13, below, on this matter. The new examples of alternation in Figure 8—figure supplement 2 and 3 illuminate the potential complexity of the prioritization mechanism. This is a topic we are currently pursuing.

(8) Given these are human subjects, could the serial nature of the decision process have to do with the subjects internally verbalising the individual decisions before combining? We noted in this context that even for eye movement responses reaction times are quite long, much longer than a monkey would take for similar tasks. How do the authors exclude the possibility of verbalisation playing a role in the time course?

The 2D RTs are long, as the reviewer says, but that’s because they are the sum of two 1D decision times (plus *T*_nd_). The 1D decision times (inferred from the fits) are normal for these types of displays, and they are similar to 1D decisions in humans on similar tasks (e.g. Bakkour et al., 2019). Of course, something like verbalization—that is, the ideation that precedes the expression of a decision (or thought)—might occur in 1D decisions too. However, there is no reason to suppose that this step would be coherence dependent, so it would be absorbed in non-decision time (see reply to 4b). It would not explain the additivity of the coherence-dependent decision times. It might help the reviewer to know that we are in the process of collecting data from monkeys. Their 1D decision times are a little faster, as is typical, but their 2D decision times also appear to be explained by a serial model. In the revised manuscript, we have modified the exposition of the non-decision time to emphasize that *T*_nd_ absorbs all coherence-independent contributions to the reaction times.

9) In the experiment with variable stimulus duration, it seems that the different durations are interleaved in a block of trials. If so, this creates additional uncertainty for the observer. Would the authors predict the same results if only one such duration was presented in a block?

Yes, we would expect the same qualitative result, but the participant might not apply the same termination criteria. The challenge, however, was to identify the one duration to use. We chose the interleaved design because it promotes the use of the same time-dependent criterion for all coherences and we wanted to compare effects on the choice functions across durations. Our approach avoids potential strategic changes of decision thresholds that could occur in a blocked design.

10) In the using a double decision with binary response:a) How can we be assured that participants can execute two independent movements at roughly the same time? We thought that bimanual coordination tasks were quite difficult to master (e.g. Mechsner et al., 2001). Could the authors comment on the amount of training required to perform their task at a reasonable motor performance level?

This comment seems to pertain to the bimanual task (not the “double decision with binary response”). The bimanual movement was facilitated by virtual channels guiding the robotic handle along each movement dimension (see Materials and methods). In fact, making the bimanual movements at roughly the same time is very easy and participants mastered the movements required for both the uni- and bimanual task within the first few blocks of training. We have added this to the Materials and methods.

b) We discovered that only correct trials were analysed in the reaction time data (last line of caption of Figure 7). Why is this an appropriate choice? Was this also the case for the other experiments?

This and the next comment pertain to the “double decision with binary response” (Experiment 5)

We thank the reviewer for pointing this out. All analyses of RT are performed on all trials except for experiment 5. In experiment 5 (same-different task) we only analyze correct trials as there is an additional challenge: We do not know which patch of dots caused the error on an error trial. That is why we resort to the *empirical* approach using γ distributions to characterize 1D decision times that would combine—by serial or parallel logic—to explain the 2D decision time, to which the *T*_nd_ is added.

In RT tasks like ours, error trials typically have longer RT than correct trials (at the same motion coherence). This is explained by a time-dependent collapsing bound, which is the normative extension of the optimal model for binary decisions (i.e., Wald’s sequential probability ratio test) when there is uncertainty about difficulty (Drugowitsch et al., 2012). Thus, we used a model with collapsing bounds in order to fit the RT on correct as well as error trials. However, for simplicity we only plot the data (and fits) for non-error trials. This is simply to reduce the number of points and fits on the graphs. By non-error, we mean the correct choices at non-zero coherence and all trials at 0% coherence. However, in Experiment 2 with very short duration stimuli we do not expect terminating bounds to play a role, so plot all trials. We also show all data in Figure 6 for reasons explained in the manuscript (see Materials and methods. Serial and parallel drift diffusion models).

c) We were wondering why the authors did not use a conjunction judgment on their original stimulus, something like "respond UP" when the stimulus is either "left-blue" or "right-yellow", and "respond DOWN" when the stimulus is either "left-yellow" or "right-blue" to "down response". Presenting two motion fields in the two hemifields seems quite different from all the other experiments.

We considered this alternative design for Experiment 5, but favored the same-different version of the binary-choice task. First, we believe that the response mapping for same vs. different is more natural and easier to learn, and it requires fewer additional computations than mapping two different color-motion conjunctions onto two response keys. This is certainly true for the subjects in Experiment 5, who had previously completed the uni- and bimanual version of the motion-color task. The alternative conjunction proposed by the Reviewers would require learning a completely new S-R mapping. Second, we believe that replicating our main findings from Experiment’s 1-4 with a different stimulus display in Experiment 5 is a strength of this experiment which we have now pointed out in the results. Also see our reply to comment 1d above.

11) Is it really reasonable to assume that evidence is retained in the buffer without any loss? If there was a loss, this would be seen in the accuracy data, so I suspect this is the reason the authors did not consider this possibility. But what is the evidence for retaining information in working memory without any loss?

The reviewer seems to have answered their own question. We reason there is a buffer and the loss of information is minimal. The only loss of information we observe is explained by ignoring information that could not be obtained because the buffer is full and its content had not been transferred yet, owing to the bottleneck. We agree that conceptualizing a buffer as a persistent representation invites the possibility of degradation, be it through decay or corruption. This is one of the motivations for the scheme proposed in Figure 8, where a sample is held in a form that is an instruction to update the decision variable, and by how much. As the reviewer notes, the lack of interference in the choice function, combined with the bookkeeping on decision time—implies there is no substantial loss. Also, we should have seen much more interference in Experiment 3—variable stimulus duration. In the conceptual model (Figure 8), the buffer can be viewed as holding an instruction; in that sense it might be compared to the time scale of holding a plan, as in an instructed memory-delay. The new Figure 8—figure supplement 2 allows for the possibility that the buffer could hold information substantially longer than 180 ms. We do not know if this is correct, but it is compatible with the class of models that ought to be considered as we forge a connection between perceptual decision-making, iconic short-term memory and dual task interference (e.g., the psychological refractory period).

12) Figure 8 is one possible instantiation of a serial bottleneck at the decision stage. But it is not clear why the transfer from the buffer has such a large delay (180ms). What is the argument against a shorter transfer time, but a longer refractory period before the decision stage can accept new information? Since this is the speculative part of the paper, perhaps it does not have to be so detailed.

We thank the reviewers for raising this concern. It demonstrates that despite our best efforts we failed to get across the rationale for the choices we made in the conceptual model (Figure 8). The following two paragraphs answer the questions. They also speak to the reason that, if anything, more detail would be desirable. In the third paragraph we describe changes to the manuscript which are intended to improve understanding of the model without exacerbating the logorrhea.

“…why the transfer from the buffer has such a large delay (180ms)” The buffer does not have a set delay. It simply cannot be emptied until the bottleneck clears, and the buffer can’t be filled until it is cleared. The limiting factor is **t**_ins_. Notice that in the 1D phase (after color has terminated in the example) the buffer is held **t**_ins_=90 ms, because after **t**_ins_ the cortex has received the ∆V instruction and clears the bottleneck to solicit another sample. This is a seamless process, which satisfies one of the desiderata guiding the exercise, namely, to maintain consistency with what is known about 1D choice-RT decisions. The 180 ms delay from buffer acquisition to release arises in the alternation phase because of the added **t**_ins_ of the color update. The motion sample is acquired when the preceding motion sample is cleared. After **t**_ins_, the cortical area that represents V_m_ clears the bottleneck, as in the 1D case, but now the color buffer releases its content, and it is another **t**_ins_ before the motion buffer can be cleared. We recognize that this is potentially confusing, and we have revised the manuscript to better communicate the ideas. See below.“…argument against a shorter transfer time, but a longer refractory period before the decision stage can accept new information” We are not sure what the reviewer means by transfer time and refractory period. Perhaps the explanation above addresses this. We are not wed to **t**_ins_=90 ms. Given latencies from display to area MT, any value greater than 60 ms would be reasonable, although the motion stimulus has another 40 ms before there is any signal. There’s also the motion filter operation which is causal and therefore adds to the delay. For the process to be non-lossy, the buffered sample must represent all of the information in the display. If the samples were acquired at shorter intervals than **t**_ins_, the buffer would not contain all the information in the stimulus. If the filters were leakier (faster time constant) the sampling at **t**_s_=**t**_ins_ would be insufficient. While it is not the main point of the current paper, this is an important insight about the temporal blurring of information in extrastriate visual cortex. We refer to Cain et al., (2013) who exploit this insight in the context of robust integration. During alternation, the sampling time is 2**t**_ins_, which leads to undersampling, as shown by the interference in Experiment 3 (variable duration) and which is compensated by obtaining more samples in the RT design.

The revised manuscript provides several brief clarifications that we hope will pre-empt the points of confusion that the reviewers experienced. We have also added three supplementary figures (to Figure 8) that demonstrate the variety of phenotypes that this scheme can exhibit. The examples provide intuition about the connection to 1D decisions, a simpler model with only one switch, like the drift diffusion model used to fit the variable duration data (Experiment 3), and probabilistic switching. In addition to furthering pedagogical goals and demystification, we think it will help readers understand why the bottleneck to perceptual decisions had escaped detection until now. If the autocorrelation times of the momentary evidence are long, temporal multiplexing would not lead to any degradation in signal to noise. The supplementary captions also serve to flesh out our choice of parameters so that an interested reader can appreciate which values were chosen in a principled manner.

13) In addition to the preference to prioritize motion, was there a tendency to prioritize the easier task, or was the motion task prioritized because observers were more sensitive to motion (at least for the eye condition)?

They were not more sensitive to motion (see our response to point 6). In the variable-duration task, where we present evidence for prioritization of motion, motion and color difficulty were adjusted individually in order to match the overall difficulty of the two dimensions. Thus, subjects were not more sensitive to motion than color in that task. Furthermore, we do not have a trial-by-trial measure of prioritization in this task that would allow us to investigate whether the easier dimension was prioritized on a given trial. Instead, prioritization in our model reflected a general tendency to integrate motion before color, regardless of task difficulty.

While we were unable to measure prioritization on a trial-by-trial basis in the variable-duration task, the bimanual task provided some interesting insights (results not shown in manuscript). First, in the bimanual task participants typically responded to the easier dimension first. Note that this does not necessarily imply prioritization, but rather, it reflects the fact that the easier dimension reached the decision bound earlier than the harder dimension. We additionally found that the coherence level on previous trials predicted which dimension participants responded to first. Specifically, participants were more likely to respond to motion first on a given trial when motion coherence was high on the previous 1-3 trials (same for color). Thus, prioritization may be streaky, suggesting that participants may apply top-down cognitive control that guides prioritization across trials, rather than prioritization being a mere bottom-up process that operates within trials. We plan to follow this up with further studies so do not focus on this observation in the current paper.